# Scaling Vision Transformers for Functional MRI with Flat Maps

Connor Lane [1 2]   Mihir Tripathy [1 3]   Leema Krishna Murali [1]   Ratna Sagari Grandhi [1]   Shamus Sim Zi Yang [1]
Sam Gijsen [1 4]   Debojyoti Das [1]   Manish Ram [1]   Utkarsh Kumar Singh [1]   Cesar Kadir Torrico Villanueva [1]
Yuxiang Wei [1 5]   Will Beddow [1]   Gianfranco Cortés [1 6]   Suin Cho [1]   Daniel Z. Kaplan [1]   Benjamin Warner [1 2]
Tanishq M. Abraham [1 2]   Paul S. Scotti [1 2]

## Abstract

We study the problem of training self-supervised foundation models for functional MRI. Our main contributions are: (1) we introduce a new model family (CortexMAE) trained using the masked autoencoder framework on 2.1K hours of open fMRI data, and (2) we release the first open evaluation suite (BRAINMARKS) for fMRI foundation models. Our core innovation is simple: we adapt the Vision Transformer to fMRI by first converting each 3D fMRI volume to a 2D map using a cortical flat map projection. We directly compare flat maps to both parcellation and volume-based representations. While each has its advantages, flat maps generally perform best. We perform the first systematic scaling analysis for fMRI and observe strict power law scaling, albeit with limits. Finally, we use BRAINMARKS to do controlled benchmark comparisons. On subject-level trait prediction, we report a challenging null result: no single model achieves clear state-of-the-art performance. Moreover, all models struggle to outperform a simple functional connectivity baseline. On cognitive state decoding, we observe more robust performance, and in this setting our CortexMAE family outperforms prior models by a large margin. Code, models, and datasets are available at https://github.com/MedARC-AI/CortexMAE and https://github.com/MedARC-AI/Brainmarks.

## 1. Introduction

A longstanding goal of neuroscience is to extract clinically useful information from functional MRI (fMRI) recordings of human brain activity (Gabrieli et al., 2015; Woo

[1]MedARC [2]Sophont [3]Baylor College of Medicine [4]University of Tübingen [5]Georgia Tech [6]University of Florida. Correspondence to: Connor Lane <contact@sophontai.com>.

*Proceedings of the 43rd International Conference on Machine Learning*, Seoul, South Korea. PMLR 306, 2026. Copyright 2026 by the author(s).

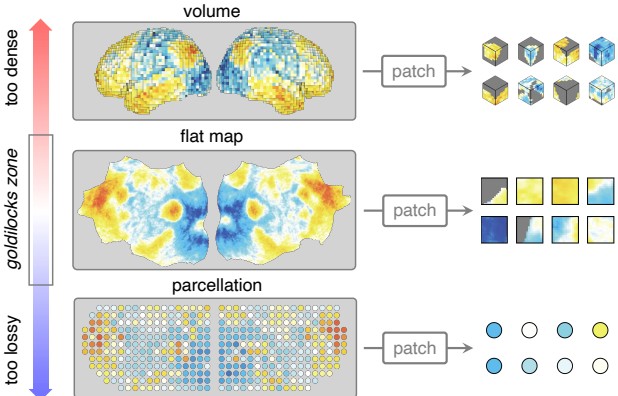

*Figure 1.* Spectrum of fMRI data representations: *volume* is the native 3D MRI format, *flat map* is our proposed representation based on cortical flat map projection, and *parcellation* data results from averaging signal within a set of regions (parcels). Volume patches are 3D cubes, flat map patches are 2D squares, and parcellation "patches" are single scalars. We hypothesize there is a "goldilocks zone" of representations that are neither too lossy nor too dense.

et al., 2017). In other domains, "foundation model" (Bommasani et al., 2021) approaches to analyzing complex medical and scientific data have made significant progress (Zhou et al., 2023; Xu et al., 2024; Wang et al., 2025b; Bodnar et al., 2025). These approaches, adapted from the broader deep learning community (Brown et al., 2020; Baevski et al., 2020; Oquab et al., 2024), combine large-scale data and compute with expressive model architectures and self-supervised learning (SSL). Can we apply the foundation model approach to unlock new applications for fMRI?

There have been many recent efforts to train self-supervised "foundation models" for fMRI data (Thomas et al., 2022; Malkiel et al., 2022; Kim et al., 2023; Caro et al., 2024; Dong et al., 2024; 2025; Wang et al., 2025a; Gijsen et al., 2025). One of the main considerations when adapting the foundation model paradigm to any new domain is how to represent the data for model input (Dosovitskiy et al., 2021). Most approaches use parcellation based representations, which reduce each 3D fMRI volume to a fixed dimension vector by averaging the activity within a set of

regions (Glasser et al., 2016; Schaefer et al., 2018). This is a computationally efficient approach with strong inductive bias from neuroscience. However, parcellating the native fMRI time series is lossy, reducing the dimensionality by $\sim 100\times$. At the other extreme, a few studies model the native 4D fMRI volume data directly (Kim et al., 2023; Wang et al., 2025a). This approach preserves the full information content of the signal and assumes no prior knowledge, but is more computationally expensive. While the *Bitter Lesson* (Sutton, 2019) reminds us that more native, agnostic approaches like this ultimately prevail, they require more data and compute to do so (Chung, 2024).

Given the current data and compute regime, we hypothesize there may be a "goldilocks zone" of intermediate fMRI representations that effectively balance these two extremes (Figure 1). To test this, we represent an fMRI time series as a series of 2D maps overlaid on a flattened cortical surface mesh (Gao et al., 2015). This flat map representation maintains the full cortical fMRI signal (like volume approaches), while reducing the raw dimensionality and injecting some inductive bias of brain geometry (like parcellation approaches). Crucially, since fMRI flat maps are standard 2D images, they are directly compatible with the standard Vision Transformer (ViT) (Dosovitskiy et al., 2021).

Using this flat map representation, we create CortexMAE: a spatiotemporal masked autoencoder (MAE-st) (He et al., 2022; Feichtenhofer et al., 2022) trained on 2.1K hours of fMRI data from the Human Connectome Project (Van Essen et al., 2013). We also train variants of CortexMAE using parcellation and volume-based representations, resulting in the first multi-representation fMRI foundation model family.

A key challenge in the fMRI foundation model field is the lack of reproducible benchmarks (Pineau et al., 2021). Although prior works use common source datasets for evaluation, the results are difficult to reproduce due to variation in dataset curation, preprocessing, and evaluation setup. To address this, we created BRAINMARKS: an open fMRI foundation model benchmark suite supporting all current models evaluated on seven public source datasets (Table 1). BRAINMARKS includes commonly reported subject-level trait prediction benchmarks, as well as dynamic cognitive state decoding benchmarks which are relatively under-studied.

CortexMAE learns to model complex spatiotemporal fMRI dynamics (Figures 3 and 4). Flat maps perform best for state decoding, while volume works best for age prediction, and parcellation is most efficient (Tables 2 and 3). We observe strict power law scaling, but with weak generalization to out-of-distribution data and a hard limit on model size (Figure 7). We do extensive ablations to analyze our model's performance (Tables 4 to 6). In a controlled benchmark (Figure 8), trait prediction performance is unreliable across all models, while state decoding is more robust. In this setting, our CortexMAE family leads all models, with the flat map variant best overall.

## 2. Related Work

**Foundation models for fMRI.** Early works exploring self-supervised learning (SSL) for fMRI include Thomas et al. (2022), Malkiel et al. (2022), and Kim et al. (2023). BrainLM (Caro et al., 2024) and Brain-JEPA (Dong et al., 2024) improve and scale up the approach, marking the first efforts to build fMRI "foundation models" (Bommasani et al., 2021). Recent extensions include Brain-Harmony (Dong et al., 2025), NeuroSTORM (Wang et al., 2025a), and Brain-Semantoks (Gijsen et al., 2025). Taken together, the works explore a broad range of modeling strategies leveraging available large-scale fMRI datasets, e.g. HCP-YA (Van Essen et al., 2013), UKBB (Miller et al., 2016). Importantly, all prior works use either parcellation or volume-based representations. Our work is the first to explore an intermediate representation (flat maps).

**Individual trait prediction** is a key application area for fMRI foundation models. The goal is to predict an individual's phenotypic traits, e.g. demographics or clinical diagnoses, invariant to within-subject variation over time. The classic approach involves fitting simple models to functional connectivity (FC) features (Finn et al., 2015; Shen et al., 2017). The approach is reminiscent of classic methods based on hand-crafted features from vision and language (Lowe, 2004; Joachims, 1998). In these other domains, moving to deep representation learning yielded immediate improvement (Krizhevsky et al., 2012). In fMRI trait prediction, however, the benefit of deep learning over simple baselines is inconclusive (He et al., 2020; Popov et al., 2024).

**Mental state decoding** is a complementary application (Kamitani & Tong, 2005; Norman et al., 2006). The goal is to predict aspects of individuals' dynamic mental state, invariant to individual differences. Specific examples include cognitive task decoding (Poldrack et al., 2009; Mensch et al., 2017; Zhang et al., 2021), reconstructing seen images (Miyawaki et al., 2008; Takagi & Nishimoto, 2023; Scotti et al., 2023; Ozcelik & VanRullen, 2023; Chen et al., 2023; Scotti et al., 2024), speech reconstruction (Défossez et al., 2023), and language reconstruction (Tang et al., 2023). A key factor for these recent advances is the public availability of large-scale fMRI datasets with rich naturalistic stimuli (Hanke et al., 2014; Chang et al., 2019; Nastase et al., 2021; Allen et al., 2022; Hebart et al., 2023). Whereas prior work has focused on task-specific models, we study how well a general-purpose "foundation model" transfers to state decoding.

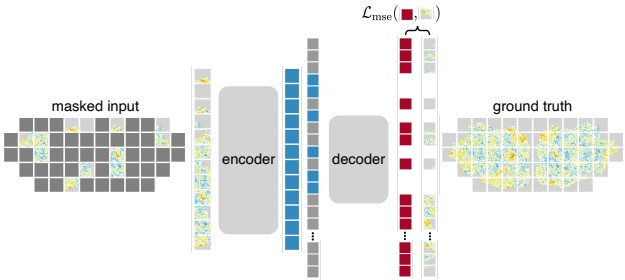

$\mathcal{L}_{\mathrm{mse}}(\blacksquare, \blacksquare)$

masked input      encoder    decoder    ground truth

*Figure 2.* MAE model architecture (He et al., 2022) adapted to fMRI flat maps. Single frame input and prediction shown for convenience. Model inputs are temporal sequences of 16 frames.

## 3. Masked Autoencoders for Functional MRI

Our approach is a straightforward adaptation of MAE-st (He et al., 2022; Feichtenhofer et al., 2022) to functional MRI (Figure 2). Briefly, an MAE consists of encoder and decoder ViTs (Dosovitskiy et al., 2021). An input image is first divided into a grid of square patches. The encoder computes embeddings for a sparse subset of observed patches, which are combined with [MASK] tokens and passed to the decoder. The two ViTs are trained jointly to minimize the mean squared error (MSE) between predicted and masked patches. After pretraining, the decoder is discarded and the encoder is applied to fully observed inputs. To extend from single images to video, the square $p \times p$ patches are simply expanded to $p_t \times p \times p$ "spacetime" patches.

**Flat map patch embedding.** To adapt MAE to fMRI, the only component we need to modify is the ViT patch embedding. To make this as straightforward as possible, we convert the 3D fMRI volumes into 2D fMRI activity *flat maps*. Data must first be preprocessed using a standard surface-based pipeline (Fischl, 2012; Glasser et al., 2013; Esteban et al., 2019). The outputs are fMRI time series mapped to a group template cortical surface mesh. We copy the surface-mapped data to a corresponding flat surface mesh from pycortex (Gao et al., 2015), and resample to a regular image grid. The resulting time series of fMRI flat maps are simply "videos" of 2D images. We can therefore use the standard spacetime ViT patch embedding directly (Arnab et al., 2021). To account for the all-zero background, we exclude entirely empty background patches and compute MSE loss only for valid, non-background pixels (Figure 2).

**Parcellation patch embedding.** For the parcellation based CortexMAE, we follow the approach of Caro et al. (2024) and Dong et al. (2024) where each parcel time series is embedded independently using a time-only patch size $p_t$. We use the Schaefer-400 parcellation (Schaefer et al., 2018), which has the same cortex-only brain coverage and results in a similar patch sequence length as the flat map approach.

**Volume patch embedding.** Volume based fMRI data can

be modeled straightforwardly using a 4D patch embedding (Hatamizadeh et al., 2022; Kim et al., 2023). However, a naive embedding of the entire 4D volume results in a $\sim 5\times$ longer patch sequence length compared to the flat map and parcellation approaches. Crucially, the relevant blood-oxygen-level-dependent (BOLD) signal is localized to only $\sim 100$K voxels of neurally active gray matter (Logothetis, 2008). We exploit this by excluding voxels outside the Schaefer cortex mask, and patch-embed the remainder using the standard 4D patch embedding. Our CortexMAE with *sparse cortical volume* patch embedding achieves a $\sim 4\times$ reduction in sequence length compared to the naive full volume strategy.

**Pretraining dataset.** We pretrain our models using openly available fMRI data from the Human Connectome Project Young Adult (HCP-YA) dataset (Van Essen et al., 2013):

| subjects | hours | runs | frames | patches |
|---|---|---|---|---|
| 980 | 2058 | 19K | 7.4M | 674M |

The dataset is made up of young adults (ages 22-35), and covers a range of experimental conditions (resting-state, 7 cognitive tasks, movie watching) with two scan protocols (3T and 7T). To account for different temporal resolutions, we resample time series to a fixed TR of 1s.

**Data normalization** is a small but important aspect of fMRI modeling. BOLD signals are in fact tiny 1-2% fluctuations in the underlying MRI image (Ogawa et al., 1990). To remove static variation due to tissue composition, we z-score normalize each voxel/ROI time series independently. To reduce global signal variation (Power et al., 2017), we also normalize each temporal frame across space. We refer to these as *coordinate* and *frame* normalization respectively.

**Implementation.** Our implementation closely follows MAE-st (Feichtenhofer et al., 2022). Model inputs are clips of 16 fMRI frames (16s duration). Our default temporal patch size $p_t$ is 4 frames. Data dimensions for the different input representations are as follows

| space | shape | dim | patches | patch size |
|---|---|---|---|---|
| parcel | $T \times 400$ | 400 | 400 | $p_t \times 1$ |
| flat | $T \times 224 \times 560$ | 77K | 364 | $p_t \times 16 \times 16$ |
| volume | $T \times 91 \times 109 \times 91$ | 132K | 465 | $p_t \times 8 \times 8 \times 8$ |

where *dim* is non-background dimensionality and *patches* is the patch sequence length. We use repeated sampling (Hoffer et al., 2020; Feichtenhofer et al., 2022) for efficient data loading. Our default masking ratio is 0.9 and we adopt tube masking (Tong et al., 2022) to prevent interpolation across time. Our default model is a vanilla MAE with a ViT-B encoder. We use a default training schedule of 625K steps with batch size 32 (512 frames). To evaluate masked reconstruction, we use a held out subset of HCP-YA subjects as well as out-of-distribution NSD data (Allen et al., 2022).

| Dataset | Target | Subjects | Samples | Seq length | TR | #Classes | Majority % |
|---|---|---|---|---|---|---|---|
| ABIDE (Di Martino et al., 2014) | ASD Dx | 578:124:124 | 578:124:124 | 150 | 2.0s | 2 | 55% |
| ADHD200 (ADHD-200, 2012) | ADHD Dx | 301:64:65 | 301:64:65 | 150 | 2.0s | 2 | 57% |
| ADNI (Jack Jr et al., 2008) | AD Dx | 328:41:41 | 328:41:41 | 100 | 3.0s | 2 | 77% |
| PPMI (Marek et al., 2011) | PD Dx | 463:99:100 | 463:99:100 | 120 | 2.5s | 2 | 62% |
| HCP-A (Bookheimer et al., 2019) | Age | 455:53:52 | 455:53:52 | 500 | 0.7s | 4 | 27% |
| HCP-A (Bookheimer et al., 2019) | Sex | 471:58:55 | 471:58:55 | 500 | 0.7s | 2 | 58% |
| HCP-YA (Van Essen et al., 2013) | Task21 | 416:88:110 | 19K:4K:5K | 16 | 1.0s | 21 | 17% |
| NSD (Allen et al., 2022) | COCO24 | 6:1:1 | 33K:5K:5K | 16 | 1.0s | 24 | 7% |

*Table 1.* Summary of trait prediction (top) and state prediction (bottom) evaluation datasets. Dx = diagnosis classification, Age = quartile classification, Task21 = cognitive task state decoding, COCO24 = object category decoding. Subject and sample counts are train:validation:test. Trait prediction datasets include one sample per subject. For diagnosis datasets, controls are the majority class.

## 4. BRAINMARKS Benchmark

To enable consistent downstream evaluation across fMRI foundation models, we built BRAINMARKS: a reproducible benchmark suite covering both subject-level trait prediction and dynamic state decoding.

**Comparison models.** We include 6 fMRI foundation models in our benchmark: SwiFT (Kim et al., 2023), BrainLM (Caro et al., 2024), Brain-JEPA (Dong et al., 2024), BrainHarmonix-F (Dong et al., 2025), NeuroSTORM (Wang et al., 2025a), and Brain-Semantoks (Gijsen et al., 2025). SwiFT and NeuroSTORM are volume-based models trained on short fMRI clips, while the others are trained on parcellated full time series. BrainHarmonix-F uses the cortex-only Schaefer-400 parcellation (Schaefer et al., 2018), while the others include subcortical structures. SwiFT is trained with contrastive learning (Dave et al., 2022), BrainLM and NeuroSTORM are trained with MAE (He et al., 2022), Brain-JEPA and BrainHarmonix-F are trained with JEPA (Assran et al., 2023), and Brain-Semantoks is trained by self-distillation (Caron et al., 2021). We also include a simple functional connectivity (FC) baseline, which uses Schaefer-400 connectome matrices as fixed feature embeddings (Hampson et al., 2006).

**Trait prediction datasets.** Following prior works (Caro et al., 2024; Dong et al., 2025), we include five datasets for predicting subject-level traits: ABIDE (Di Martino et al., 2014) for Autism (ASD) classification, ADHD-200 (ADHD-200, 2012) for ADHD classification, ADNI (Jack Jr et al., 2008) for Alzheimer's Disease (AD) classification, PPMI (Marek et al., 2011) for Parkinson's disease classification, and HCP-A (Bookheimer et al., 2019) for age and sex classification (Table 1, top). For ADHD-200, we combine across ADHD subcategories. For PPMI, we use prodromal subjects as controls for better class balance. For consistency with the other datasets, we formulate HCP-A age prediction as classification by discretizing into four quartile bins. For each dataset, we construct a curated subset of 400-900 subjects, including one 5-7 minute resting-state fMRI run per subject.

**State prediction datasets.** We include two datasets for decoding a subject's dynamic cognitive state: HCP-YA task-state prediction (Task21) (Van Essen et al., 2013), and NSD COCO object category decoding (COCO24) (Allen et al., 2022) (Table 1, bottom). The HCP-YA Task21 benchmark follows the setup of prior works (Zhang et al., 2021; 2022; Rastegarnia et al., 2023). The task is to classify which of 21 cognitive conditions a subject is in (e.g. story listening, finger tapping) based on a short 16s fMRI clip.

NSD COCO24 is a visual category decoding benchmark similar to those used in prior works (Horikawa & Kamitani, 2017; Chang et al., 2019; Chen et al., 2023). We use CLIP (ViT-L/14) (Radford et al., 2021) to assign each NSD stimulus image a global COCO category label. We exclude ambiguous images (CLIP confidence $<0.9$), and categories with too few remaining examples ($<600$), leaving 25K images from 24 highly distinct categories (e.g. "motorcycle", "zebra", "pizza"). The task is to decode the seen object category from a 16s fMRI clip time-locked to stimulus image onset. Due to the short 4s trial duration in NSD, each fMRI clip contains overlapping responses to multiple image presentations. Models must therefore learn to attend to the target response while ignoring the others.

To focus on general cross-subject decoding, the validation and test splits for both state prediction datasets are constructed with held out subjects. For HCP-YA, the test subjects are also excluded from the HCP-YA pretraining set. For NSD, the held out splits also use unseen images.

**Evaluation setup.** We use a probe based evaluation following standard practice in vision SSL (Balestriero et al., 2023). For trait prediction, we use a simple and reliable linear probe setup to handle the small sample sizes. We train logistic regression classifiers (Pedregosa et al., 2011) on average-pooled embeddings and report average performance over 100 randomized train-test splits. Since state prediction datasets have more samples, we are able to use a more sensitive approach. We train attentive probe classifiers (Assran et al., 2023; Darcet et al., 2025) on unpooled

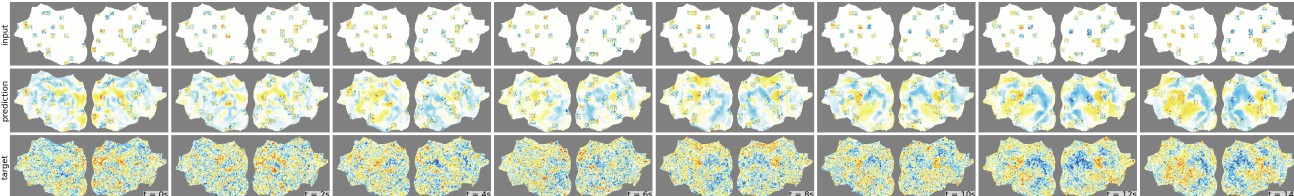

*Figure 3.* MAE predictions on fMRI flat maps. We show the masked input (top), prediction (middle), and target (bottom) for 8 frames spaced 2s apart from left to right. RGB color mapping for visualization; model inputs are single channel.

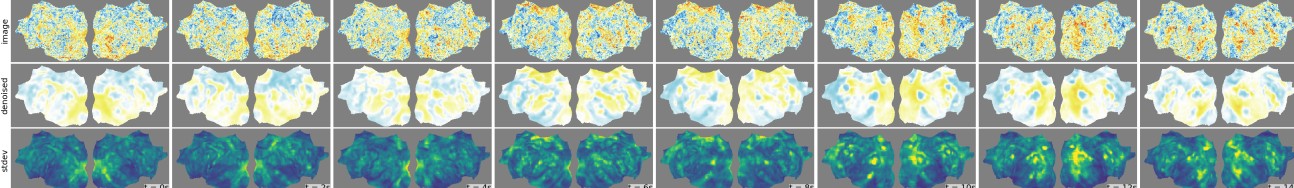

*Figure 4.* MAE denoising on fMRI flat maps. We show the original image (top), denoised prediction (middle), and standard deviation maps (bottom). The prediction is computed by averaging the masked reconstruction over 100 mask samples (excluding predictions for observed patches). The standard deviation maps capture how predictions vary depending on observed context.

embeddings and report performance for the single fixed split. All models are tuned with the same protocol. For trait prediction, we use 5-fold cross-validation with the default scikit-learn hyperparameter grid. For state prediction, we tune the attentive probe learning rate independently for each model over a dense grid of 49 values (Darcet et al., 2025) and apply early stopping. We use probes rather than fine-tuning to isolate the effect of pretraining and to keep evaluation cheap enough to apply uniformly to all models.

## 5. Experiments

Figure 3 shows the masked reconstruction of our default flat map CortexMAE for an HCP-YA subject not seen during pretraining. Our model is able to reconstruct precise fMRI activity patterns given limited context.

In Figure 4, we apply our model to fMRI denoising. For a given input, we generate 100 reconstructions conditioned on different random masks and average the predictions. The denoised reconstructions recover the large-scale spatiotemporal dynamics of the input. Unstructured noise is unpredictable and is left behind. This highlights how models like CortexMAE can offer a new approach to fMRI denoising, by leveraging complex population priors learned from large-scale data (Elad et al. 2023; but see also Kay 2022).

Figure 5 shows how the first principal component of the model's spatial position embedding mirrors the brain's default mode network (Raichle, 2015), represented here as the FC principal gradient (Margulies et al., 2016). While some prior works design position embeddings to explicitly encode functional network structure (Dong et al., 2024; Gijsen et al., 2025), our model shows that this structure can also emerge naturally during training.

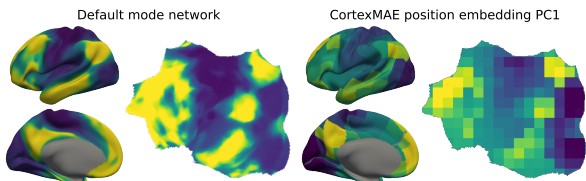

Default mode network     CortexMAE position embedding PC1

*Figure 5.* CortexMAE learns the brain's default mode network from scratch. (left) Map of default mode network (principal gradient, Margulies et al. 2016). (right) First principal component of the model's learned spatial position embedding.

### 5.1. Comparison of fMRI Input Representations

In this section, we compare the three CortexMAE variants trained with different representations: parcel, flat map, and cortical volume. To have a reliable comparison, we repeat pretraining 8 times with different random seeds. Figure 6 visualizes reconstructions for a common example. The parcel and volume reconstructions are projected to the flat map for consistent visualization. All models capture similar aspects of the signal. The flat map model's targets and predictions are more detailed than the parcel, yet more structured and less noisy than the volume.

**Downstream probe comparison.** Table 2 reports the downstream probe performance for each model, averaged over 8 pretraining repeats. For the clinical diagnosis datasets (ABIDE, ADHD200, ADNI, PPMI), we observe no reliable differences between input representations, likely reflecting the small sample sizes (Table 1). We discuss cross-model comparison on these datasets in Section 5.4.

On age prediction, the volume model reliably outperforms the other two models. This may be driven by structural features such as age related cortical thinning

| space | ABIDE | ADHD200 | ADNI | PPMI | HCP-A Age | HCP-A Sex | HCP-YA Task21 | NSD COCO24 |
|---|---|---|---|---|---|---|---|---|
| parcel | 62.0 $\pm 0.8$ | 56.8 $\pm 0.6$ | 61.6 $\pm 1.2$ | 61.4 $\pm 1.3$ | 44.2 $\pm 0.5$ | 71.2 $\pm 1.0$ | 97.5 $\pm 0.2$ | 27.5 $\pm 0.5$ |
| flat | 61.4 $\pm 1.3$ | 59.2 $\pm 1.0$ | 62.4 $\pm 1.4$ | 58.8 $\pm 1.1$ | 47.5 $\pm 1.6$ | **87.4** $\pm 0.7$ | **98.9** $\pm 0.1$ | **31.0** $\pm 0.7$ |
| volume | 60.4 $\pm 0.8$ | 58.8 $\pm 1.1$ | 64.3 $\pm 1.6$ | 59.1 $\pm 1.2$ | **53.4** $\pm 0.5$ | 86.3 $\pm 0.7$ | 96.2 $\pm 0.3$ | 27.7 $\pm 0.7$ |
| connectome | 59.8 | 57.0 | 58.6 | 58.0 | 45.6 | 81.9 | 82.4 | 7.4 |

*Table 2.* Downstream probe performance across fMRI representations. Values are mean accuracy $\pm$ standard deviation over 8 random repeats. **Bold** indicates *robust* improvement over non-bold ($p < 0.0001$). The flat map MAE performs best on dynamic state classification. The volume MAE performs best on age classification. Both volume and flat outperform parcel on sex classification.

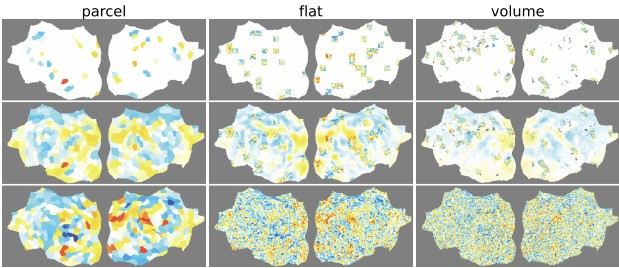

*Figure 6.* MAE predictions across different fMRI representations for a single example. We project the parcel and volume data to the cortical flat map for consistent visualization. This projection is lossy for the volume model, capturing only voxels intersecting the cortical surface.

| space | time | params | FLOPs | compute | data |
|---|---|---|---|---|---|
| parcel | 11 hr | 85M | 89G | 10K fps | 60K fps |
| flat | 28 hr | 86M | 92G | 9K fps | 4K fps |
| volume | 50 hr | 87M | 116G | 8K fps | 2K fps |

*Table 3.* Training time comparison across representations for default setting (ViT-B) on a single H100. Parameter count is for encoder only. FLOP count is for forward pass on a single sample. Compute and data loading are in fMRI frames per second (fps).

(Bethlehem et al., 2022) leaking into the dense volume-based fMRI representation. Regardless, it is a positive sign for future work toward dense fMRI models.

Our proposed flat map model shows a clear advantage for dynamic state prediction (Table 2, right). These results support our hypothesis that fMRI modeling benefits from an intermediate "goldilocks" representation.

**Compute comparison.** Table 3 analyzes the compute cost of the different models. The models use the same architecture (ViT-B) and similar number of patches (364-465), so the parameter and FLOP counts are roughly similar. Nonetheless, the parcel MAE is 2.5× faster to train than the flat map MAE, which in turn is 1.8× faster than volume. For the dense flat map and volume models, training is bottlenecked by data loading, while the parcel model is compute bound. Our volume MAE achieves significant compute and IO savings over prior volume models by restricting to cortical gray matter (132K voxels) rather than processing the full MRI volume (900K voxels). The flat map MAE is even more efficient, due to its more compressed, structured representation. Future work may continue to explore more efficient dense fMRI representations.

### 5.2. Scaling laws for fMRI

**Scaling with dataset size.** To analyze how masked reconstruction scales with the amount of pretraining data, we emulate the approach of Kaplan et al. (2020) and Hoffmann

et al. (2022). Similar to language modeling, the grounded masked prediction objective of MAE provides a natural setting for scaling analysis (Xie et al., 2023). We pretrain our default flat map CortexMAE (ViT-B) on varying size subsets of HCP-YA from 400K frames (110 hours, 50 subjects) to 6.6M frames (1.8K hours, 880 subjects).

Figure 7a visualizes MAE loss on the HCP-YA and NSD validation sets. For the in-distribution held-out data from HCP-YA, we find that the test loss decreases with increasing training dataset size according to a strict power "scaling law" (Figure 7a, top). This mirrors classic data scaling results in language modeling (Kaplan et al., 2020; Hoffmann et al., 2022)[1]. This scaling behavior does not perfectly carry over to the out-of-distribution NSD validation set, however. As with HCP-YA, reconstruction on NSD improves with dataset size. But the rate of improvement slows compared to the power law prediction (Figure 7a, bottom). This raises the possibility that greater dataset *diversity*, as well as scale, is required for more generalizable representations.

**Scaling with model size.** Figure 7b shows a similar scaling analysis over model size. To vary model size, we scale the encoder depth of our CortexMAE from 3 to 15 while keeping model proportions (depth-to-width ratio, encoder-to-decoder depth ratio) matched to the default ViT-B encoder (Tan & Le, 2019; Hoffmann et al., 2022; Karpathy, 2025). We use the same training schedule and hyperparameters for all models. Similar to data scaling, MAE reconstruction improves with model size. However, the improvement saturates at depth 9 (37M encoder parameters) for both HCP-YA

---

[1]The exponent is -0.01 vs -0.1 in Kaplan et al. (2020), however, indicating weaker scaling than for next token prediction.

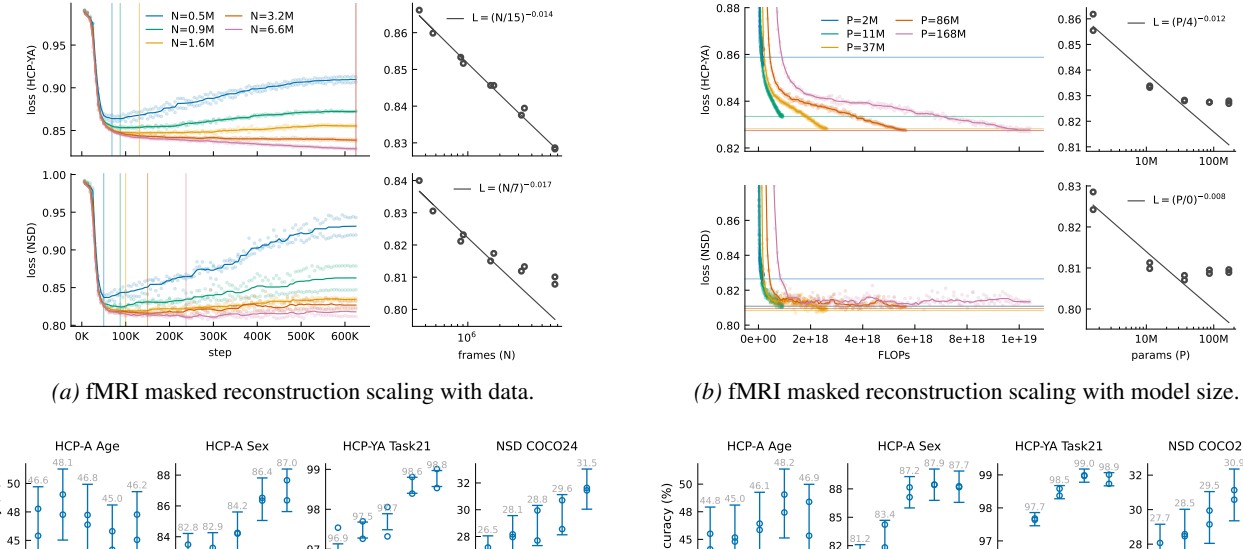

*(a)* fMRI masked reconstruction scaling with data.

*(b)* fMRI masked reconstruction scaling with model size.

*(c)* Downstream probe accuracy scaling with data.

*(d)* Downstream probe accuracy scaling with model size.

*Figure 7.* Scaling analysis. (a) Models are trained on subsets of HCP-YA (split by subject). Lines indicate rolling median over 5 epochs. Vertical lines indicate best epochs for each dataset. Power laws are estimated from best losses using the three smallest data splits. (b) Encoder depth is scaled from 3 to 15 while fixing model proportions. Horizontal lines indicate best loss for each model. Power laws are estimated from best losses using the three smallest models. (c-d) Error bars indicate $\pm 2 \times$stdev, using the standard deviations in Table 2.

and NSD. This suggests that a relatively small capacity is sufficient to model all of HCP-YA.

**Effects of scale on downstream prediction.** Figures 7c and 7d shows that downstream prediction performance scales reliably with dataset and model size. We report probe accuracy across four downstream datasets spanning trait and state prediction. We observe scaling trends consistent with reconstruction loss scaling in all datasets except HCP-A age. On NSD COCO24 for example, the largest data scale outperforms the smallest by ∼5%. As with reconstruction loss, downstream prediction performance begins to saturate around 37M parameter model size.

### 5.3. Ablation Experiments

In this section, we analyze our model's performance in a series of ablation experiments. We take advantage of the flat map representation to directly apply several techniques developed from images and video.

**Mask sampling.** The mask sampling strategy and ratio are arguably the most important components of an MAE model. In Table 4a, we compare standard uniform masking (He et al., 2022), our default tube masking (Tong et al., 2022), and block tube masking with $2 \times$ patch blocks ($32 \times 32$ patches) (Yang et al., 2025). Tube masking prevents local interpolation across time, while block tube masking promotes longer range prediction. All masking strategies

perform well. Uniform masking requires a higher masking ratio, while block tube masking struggles at the highest ratio.

**Data augmentation.** Data augmentation is an underexplored area for fMRI modeling. We evaluate several simple augmentation methods (Table 4b): temporal TR scaling, where we randomize the TR in $[0.8, 1.25]$, grayscale jitter, weak random crop with fixed aspect ratio, and strong random crop with random aspect ratio. None of the augmentations result in robust improvements over baseline, although TR scaling appears to have a modest effect. This suggests that natural image augmentations like cropping are not a good fit for fMRI. Developing useful data augmentations specific for fMRI remains an open problem.

**Reconstruction target.** He et al. (2022) found that focusing MAEs on higher frequency content by z-score normalizing each target patch ("patch norm") improves performance. Like natural images, fMRI data are dominated by low spatial frequency signal. Moreover, the low frequency structure in fMRI can be characterized more strongly: most of the signal is explained by just a few stereotypical components (Margulies et al., 2016; Bolt et al., 2022). We extend the idea of patch normalization to account for this. Specifically, we orthogonalize target frames with respect to the first few frame-wise principal components ("PC norm", see also Rodriguez et al. 2025). In Table 4c, patch norm and PC norm

| strategy | 50 | 75 | 90 | 95 |
|---|---|---|---|---|
| uniform | 25.7 | 28.1 | 29.8 | 29.7 |
| tube | 23.4 | 29.9 | 31.4 | 30.4 |
| tube (2×) | 26.6 | 31.1 | 30.6 | 28.7 |

*(a)* Mask sampling across ratio.

| case | acc. |
|---|---|
| none | 30.4 |
| TR scale | 31.7 |
| gray jitter | 29.7 |
| crop (weak) | 28.9 |
| crop (strong) | 24.4 |

*(b)* Augmentation.

| case | acc. |
|---|---|
| patch | 29.5 |
| patch norm | 31.4 |
| PC norm ($d$=2) | 32.1 |
| PC norm ($d$=8) | 24.6 |

*(c)* Reconstruction target.

| $p_t$ | patches | acc |
|---|---|---|
| 16 | 364 | 27.6 |
| 8 | 728 | 28.3 |
| 4 | 1456 | 29.6 |
| 2 | 2912 | 32.9 |
| 1 | 5824 | 31.9 |

*(d)* Temporal patch size

*Table 4.* Ablations on NSD COCO24. The model is our default flat map CortexMAE (ViT-B). Default settings in gray . red indicates $> 3\sigma$ below baseline. No scores $> 3\sigma$ above baseline. (a) All masking strategies perform well. Uniform requires higher ratio. tube 2× (Yang et al., 2025) requires lower ratio. (b) No clear benefit from any augmentation. (c) No clear benefit from target patch normalization (He et al., 2022) or global PCA normalization. (d) Better performance for smaller temporal patch size.

| parcellation | Age | Sex | Task21 | COCO24 |
|---|---|---|---|---|
| S400 | 44.1 | 70.8 | 97.3 | 27.5 |
| S400 + Tian S3 | 43.5 | 72.5 | 97.5 | 27.2 |
| A424 | 44.3 | 71.4 | 96.7 | 26.0 |

*Table 5.* Subcortical structures have little effect. We compare parcellation CortexMAE models with Schaefer-400 (cortex only), Schaefer-400 + Tian S3 (cortex + subcortex), and A424 (cortex + subcortex + cerebellum). red indicates $> 3\sigma$ below baseline .

| global | coord | frame | Age | Sex | Task21 | COCO24 |
|---|---|---|---|---|---|---|
| ✓ | ✓ | ✓ | 48.1 | 87.6 | 98.8 | 31.4 |
| ✓ | ✓ | ✗ | 50.8 | 87.3 | 97.7 | 26.3 |
| ✓ | ✗ | ✗ | 40.2 | 79.1 | 16.1 | 5.5 |
| ✓ | eval | ✗ | 40.8 | 74.5 | 74.9 | 9.5 |

*Table 6.* Input normalization is essential for state decoding. global = global normalization. coord = per-coordinate time series normalization. frame = per-frame spatial normalization. Bottom row uses coord norm during evaluation but *not* pretraining.

with two components result in modest but non-significant improvements, while more aggressive PC norm hurts prediction. Developing better techniques to focus fMRI models on fine-grained detail is another open problem.

**Temporal patch size.** Scaling the model's token capacity by reducing the temporal patch size $p_t$ results in consistent performance improvements up to size 2 (Table 4d). This suggests that as with standard ViTs, there is a speed/accuracy tradeoff for smaller patches (Beyer et al., 2023). However, we do not observe a similar effect for smaller spatial patch size (Table 10).

**Subcortical structures.** A key limitation of the flat map representation is that it does not directly support subcortical structures, which are key nodes in the brain's functional network (Park et al., 2024). To estimate the impact of excluding subcortex, we evaluate parcellation based CortexMAE models pretrained with three parcellations: Schaefer-400 (Schaefer et al., 2018) (cortex only, default), Schaefer-400 + Tian S3 (Tian et al., 2020) (cortex + subcortex, used by Brain-JEPA), and A424 (Nemati et al., 2020) (cortex + subcor-

tex + cerebellum, used by BrainLM). Including subcortical structures does not improve model performance (Table 5). State decoding performance using A424 is worse than the Schaefer-400 baseline. A424 is derived from the Glasser parcellation (Glasser et al., 2016), which uses less spatially compact parcels compared to Schaefer.

**Input normalization.** Finally, we look at how different choices for input normalization affect performance. Our default setup applies both coordinate normalization across time, and per-frame normalization across space. Removing these steps results in dramatic loss of performance on state prediction (Table 6). In particular, without coordinate normalization performance is near chance. The model is effectively *blind* to the functional part of the fMRI signal. If we apply coordinate normalization at evaluation time to models pretrained without it (Table 6, bottom row), state prediction performance increases above chance, but not to the level of models trained with the normalization from scratch. Interestingly, performance on trait prediction is not affected in the same way. Even the model trained with only global normalization achieves competitive performance. Subject-level trait prediction can use static as well as dynamic features, while state prediction requires dynamics.

**Supplementary ablations.** Appendix C includes more ablations testing the effects of temporal sequence length (Table 9), decoder depth (Table 11a), drop path rate (Table 11b), and probe depth (Table 11c). We also test the effects of pretraining on resting-state vs task fMRI (Table 12) and larger scale pretraining on UKBB (Miller et al., 2016) (Figure 11).

### 5.4. Benchmarking Against Prior Work

Here we compare our models (CortexMAE-{P,F,V} for parcel, flat map, volume respectively; Table 2) against prior models using the BRAINMARKS benchmark suite.

**Trait prediction comparison.** Overall, we observe inconsistent performance on trait prediction benchmarks (Figure 8). On the four clinical diagnostic datasets (ABIDE, ADHD200, ADNI, PPMI), the foundation models struggle to reliably outperform the simple FC baseline. The mod-

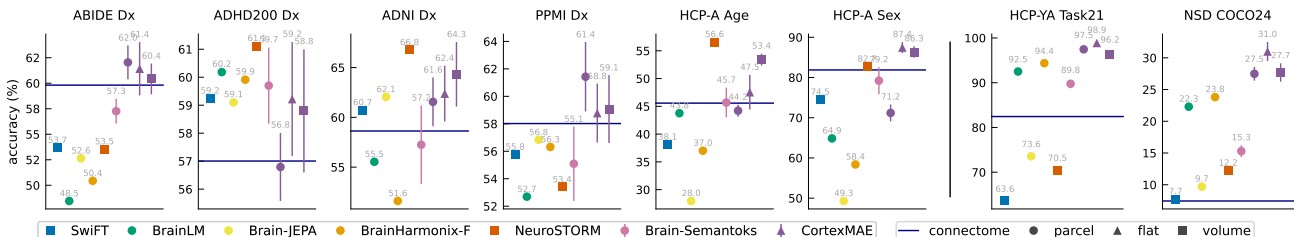

*Figure 8.* fMRI foundation model probe comparison for subject-level trait prediction (left; ABIDE, ADHD200, ADNI, PPMI, HCP-A) and dynamic state prediction (right; HCP-YA Task21, NSD COCO24). To handle small sample size and class imbalance, trait prediction uses balanced accuracy and logistic probe over 100 random splits. State prediction uses raw accuracy and attentive probe over a single fixed split. Confidence intervals indicate $\pm 2\times$stdev over pretraining repeats (only available for CortexMAE and Brain-Semantoks).

els appear poorly differentiated, and model ranking varies across datasets. On HCP-A age and sex classification, we observe somewhat more robust differences. On age classification in particular, NeuroSTORM and CortexMAE-V outperform the baseline and all other models. Both are volume-based models and may be sensitive to structural brain changes during aging (Bethlehem et al., 2022). Importantly, CortexMAE-V was trained only on HCP-YA (ages 22-35) and saw no examples from the HCP-A age range (36-100+) during pretraining. Whereas the pretraining dataset for NeuroSTORM includes subjects specifically from HCP-A, as well as other older adults (Wang et al., 2025a).

To our knowledge, this is the first benchmark highlighting inconsistent performance of fMRI foundation models on trait prediction. We have taken significant steps to support a controlled, fair evaluation (Section 5.4, Appendix B.4). However, there are of course limitations of our current evaluation. For example, we do not currently implement model-specific nuisance regression strategies. Importantly, our benchmark is open-source and fully reproducible. We invite the community to collaborate with us toward more robust fMRI foundation model evaluation.

**State prediction comparison.** On dynamic state prediction, by contrast, we observe more robust performance. The model ranking is consistent across both datasets, and most models outperform the simple FC baseline. At the same time, our CortexMAE models robustly outperform all other models. The parcellation and volume based models are the best in their respective input representation classes, while the flat map model performs best overall.

Some models (SwiFT, Brain-JEPA, NeuroSTORM) appear to have been pretrained without coordinate normalization, which likely explains their poor performance on these tasks (Table 6)[2]. The performance of Brain-Semantoks vs other parcellation models may be due to its coarse (but compute efficient) network-based tokenization.

We hope these results motivate future work to focus more on dynamic mental state prediction. Due to much larger sample sizes (Table 1), state decoding benchmarks are able to measure performance more reliably than trait prediction. But more importantly, state based evaluations are able to distinguish function-specific fMRI foundation models vs models that are largely sensitive to the underlying structural part of the fMRI image.

## 6. Conclusion

Our goal in this work was to do a straightforward study of how to train an fMRI foundation model. We created CortexMAE: a family of fMRI foundation models trained with vanilla MAE-st on 2.1K hours of open fMRI data from HCP-YA. Our flagship model based on a simple flat map representation achieves SotA performance on cognitive state decoding, while our sparse cortical volume model performs well on age prediction, and the parcellation based model is most efficient. Our results provide initial support for a "goldilocks zone" hypothesis: the best fMRI representations (given current data and compute) should be neither too structured, nor too dense. The fact that flat maps do not universally outperform the alternatives, however, suggests that there is still room for improvement. We establish the first rigorous scaling laws for fMRI, while at the same time highlighting the limits of scaling with homogeneous data. Finally, we created BRAINMARKS: an open, reproducible benchmark suite. Our benchmark comparison shows that fMRI foundation models significantly outperform baselines on dynamic cognitive state prediction, while at the same time failing to do so on standard trait prediction benchmarks.

Key limitations to explore in future work include: (1) developing even more effective intermediate representations of fMRI data, (2) scaling pretraining beyond single-source datasets, (3) expanding and continuing to standardize the set of fMRI foundation model benchmarks, and (4) exploring ways to leverage models' unique representations of dynamic brain state for clinical application.

---

[2]Following Table 6, the models are still *evaluated* with coord norm enabled. Otherwise, their performance is near chance (Table 15).

## Impact Statement

Functional MRI has found much less clinical application compared to other medical imaging modalities (Gabrieli et al., 2015). Partly this is due to intrinsic limitations of the modality, such as low SNR, low spatiotemporal resolution, and loose coupling between neural firing and BOLD activity (Constable, 2023). Another aspect of the challenge, however, is that the data are significantly *out-of-distribution* with respect to our everyday visual experience. This makes it impossible for a radiologist to look at a raw fMRI time series and make sense of it. Foundation models could become a kind of perceptual "prosthesis" for interpreting fMRI data. By training models to natively *see* fMRI, and then analyzing their representations in turn, we could unlock broad new applications under the nascent field of functional neuroradiology (Faro et al., 2011).

At the same time, there are important potential ethical concerns (Rainey et al., 2020). Decoding aspects of a person's mental state from fMRI raises privacy concerns. However, current fMRI decoding methods are low fidelity and require cooperation from the participant. Our approach to mitigate ethical risk is to do research *in the open*, using open data as much as possible.

## Acknowledgements

Thanks to FAL AI for providing compute that supported this research. Thanks to MedARC contributors Melvin Selim Atay, Mohammed Baharoon, Atmadeep Banerjee, Uday Bondi, Pierre Chambon, Alexey Kudrinsky, Souvik Mandal, Ashutosh Narang, Alex Nguyen, Yashvir Sabharwal, Kevin Son, and Dingli Yu for contributing to an earlier version of this project. Thanks to the MedARC Discord community in general for being the public forum from which this research was developed. Thanks to Zijiao Chen, Gregory Kiar, and Florian Rupprecht for helpful discussions on an earlier version of this work. Thanks to the anonymous reviewers for helpful feedback.

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

# A. Open Research: 100% Transparent Volunteer-Driven Science

CortexMAE and BRAINMARKS were openly developed through volunteer contributions in the MedARC Discord server. Source code was always accessible via a public GitHub repository throughout the lifespan of the projects. Research discussions were held via public Discord channels, and weekly video conference calls were recorded and shared publicly. We continue to extend a global invitation to contribute to MedARC projects to cultivate an internationally diversified, volunteer-driven research team. We contend that fully transparent open-research initiatives could redefine the traditional framework of scientific research, democratizing entry into machine learning and medical research through the harnessing of crowd-sourced collective intelligence and community collaboration.

## A.1. Author Contributions

**CL** project lead. **MT** scaling experiments, dataset curation for ADNI and HCP-A, BrainLM integration, UKBB pretraining. **LKM** comparison of preprocessing pipelines, streaming pretraining experiments, dataset curation for ABIDE. **RSG** dataset curation for PPMI, PCA feature visualization. **SSZY** masking strategy and mask ratio experiments, Brain-JEPA model integration. **SG** Brain-Semantoks model integration, manuscript review. **DD** decoder architecture experiments, NSD CLIP regression evaluation. **MR** masking strategy implementation, pretraining smoke test implementation. **UKS** CAPI pretraining experiments. **CKTV** SwiFT and NeuroSTORM model integration. **YW** experiments with "vector parcel embedding" ViTs. **WB** BrainHarmonix-F integration. **GC** MLP baseline implementation. **SC** data augmentation implementation. **DZK** project feedback and manuscript review. **BW** project feedback and manuscript review. **TMA** project supervisor. **PSS** project supervisor, senior investigator.

# B. Additional methods

## B.1. Flat map construction

We use the precomputed `fsaverage` flat map distributed with pycortex (Gao et al., 2015), which we resample onto the `32k_fs_LR` template mesh using the connectome workbench (Marcus et al., 2013; Glasser et al., 2013). We exclude invalid medial wall vertices (which have a non-zero $z$ component in flat map coordinates), and intersect with the Schaefer parcellation mask (Schaefer et al., 2018) to yield a valid flat map mask containing 58212 vertices across both cortical hemispheres. We fit a regular grid of size height $\times$ width $= 224 \times 560$ to the array of $(x, y)$ points contained in the mask. The grid has a pixel resolution of 1.2mm in flat map coordinates, which equals the mean nearest neighbor

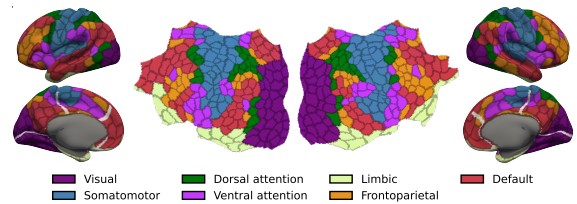

| | | | |
|---|---|---|---|
| ■ Visual | ■ Dorsal attention | ■ Limbic | ■ Default |
| ■ Somatomotor | ■ Ventral attention | ■ Frontoparietal | |

*Figure 9.* Schaefer 400 parcellation (Schaefer et al., 2018) with Yeo resting-state networks (Yeo et al., 2011) on the cortical surface and flat map. Relaxation cuts required for flat map transformation (Gao et al., 2015) are marked in white.

distance. To project surface-mapped fMRI data onto the flat map grid, we extract the array of values corresponding to our flat map vertex mask and then resample using linear interpolation (Virtanen et al., 2020). After resampling, there are 77763 pixels contained in the flat map mask. Figure 9 shows the correspondence between surface and flat map space using the Yeo resting-state networks overlaid on the Schaefer 400 parcellation (Yeo et al., 2011).

## B.2. Dataset Preprocessing

The HCP-YA, HCP-A, NSD, and UKBB datasets use the official preprocessed data derivatives provided by the collecting institutions. The HCP-A dataset preparation uses ICA-FIX denoised outputs (Salimi-Khorshidi et al., 2014), whereas the HCP-YA preparation uses data without any nuisance regression. ABIDE, ADHD-200, ADNI, and PPMI were preprocessed using fMRIPrep v25.2.3 (Esteban et al., 2019). No nuisance regression was applied. Flat map and parcellation models use preprocessed outputs in CIFTI "grayordinate" fsLR 91K space, whereas volume models use MNI152 2mm (FSL NLin6Asym) outputs.

| config | value |
|---|---|
| optimizer | AdamW |
| momentum | $\beta_1, \beta_2 = 0.9, 0.95$ |
| weight decay | 0.05 |
| learning rate | 1.25e-4 (flat, vol), 3.75e-5 (parcel) |
| lr schedule | cosine decay |
| warmup steps | 31K |
| total steps | 625K |
| batch size | 32 |
| gradient clipping | 1.0 |

*Table 7.* Pretraining setting on HCP-YA.

## B.3. Pretraining implementation details

The default pretraining config is in Table 7. We use the AdamW optimizer (Loshchilov & Hutter, 2017) and cosine learning rate decay (Loshchilov & Hutter, 2016). In total, the model sees 320M fMRI frames during pretraining, which is $\sim$43 effective epochs over our HCP-YA training set. We use linear learning rate scaling (Goyal et al., 2017) ($\texttt{lr} = \texttt{base\_lr} \times \texttt{batch\_size}/256$). We tuned the base

| config | value |
|---|---|
| optimizer | AdamW |
| momentum | $\beta_1, \beta_2 = 0.9, 0.999$ |
| base learning rate | 3e-4 |
| base weight decay | 0.05 |
| lr scale grid | [0.02, 0.023, 0.028, 0.033, 0.038, 0.045, 0.053, 0.062, 0.074, 0.087, 0.1, 0.12, 0.14, 0.17, 0.2, 0.23, 0.27, 0.32, 0.38, 0.44, 0.52, 0.61, 0.72, 0.85, 1, 1.2, 1.4, 1.6, 1.9, 2.3, 2.7, 3.1, 3.7, 4.3, 5.1, 6, 7.1, 8.3, 9.8, 12, 14, 16, 19, 22, 26, 31, 36, 43, 50] |
| wd scale grid | [1.0] |
| batch size | 128 |
| total steps | 4000 |
| warmup steps | 1000 |
| lr schedule | cosine decay |
| early stop period | 200 |

*Table 8.* Attentive probe setting

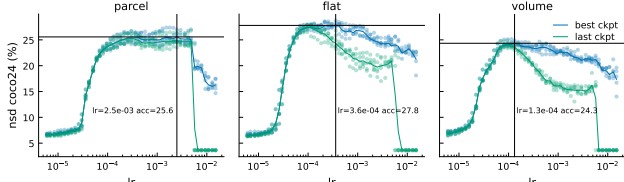

*Figure 10.* Attentive probe performance across learning rate for best and last checkpoints. The dense learning rate grid (trained in parallel) allows precise tuning of all models.

lr for each input representation separately, resulting in values 1e-3 for flat and volume and 3e-4 for parcellation. We use repeated sampling (Feichtenhofer et al., 2022; Hoffer et al., 2020) to improve data loading throughput. Each time an fMRI run is loaded from disk, we extract $4 \cdot N_t/16$ random clips, where $N_t$ is the length of the run. The clips are then appended to an in-memory shuffle buffer, which we sample from to construct training batches using WebDataset.

Our default model is a vanilla MAE (He et al., 2022; Feichtenhofer et al., 2022) with some minor changes. To prevent large initial loss and gradient spikes, we initialize the decoder head weights to zero (Beyer et al., 2023). We remove redundant position embeddings added to the learned [CLS] token, and we remove decoder position encoding from the encoded embeddings, since they already contain position information. We train with mixed precision at float16. We observe significant training instability with bfloat16 (especially without the zero-init of the decoder head).

### B.4. Model comparison protocol

**Trait prediction.** Subject-level trait prediction performance is evaluated using a logistic regression probe on top of frozen embeddings. We use the average-pooled embedding across patches for all models. Although some models (Brain-Semantoks, CortexMAE) expose a [CLS] embedding, they do not predict better. The classifier is scikit-learn LogisticRegressionCV with default hyperparameters, standard-scale preprocessing, and 5-fold internal cross-validation. To reduce variance, we average performance over 100 repeats with randomized train/test splits, stratifying by target to preserve class proportions. We report balanced accuracy to account for class imbalance.

**State prediction.** Trial-level state prediction is evaluated using the more performant attentive probe (Assran et al., 2023) (which would be compute-prohibitive in the noisy trait pre-

diction setting). The config used for all models is in Table 8. We train parallel attentive probes over a grid of learning rates following (Darcet et al., 2025) and choose the best by validation accuracy. We also use early stopping (i.e. select the best performing checkpoint by validation accuracy). We find that attentive probe performance is sensitive to learning rate and training schedule, but relatively robust to changes in other parameters like weight decay. We use a dense learning rate scale grid of 49 values (`np.logspace(-1.7, 1.7, 49)`) for precise tuning of all models (Figure 10).

**Sequence length.** We resample inputs to each model's target TR using linear interpolation (with a tolerance of 0.1 s). For inputs shorter than a model's expected temporal sequence length, we pad the input with the per-coordinate mean. For inputs longer than expected, we apply the model on non-overlapping sliding windows.

## C. Additional experiments

### C.1. Supplementary ablations

**Input sequence length.** Existing parcellation based models like BrainLM (Caro et al., 2024) and Brain-JEPA (Dong et al., 2024) use long input time series (>2 min). This enables them to model long temporal dependencies, possibly at the expense of fine-grained state representation. Table 9 shows that pretraining with longer input time series hurts HCP-YA and NSD state decoding. This drop likely explains much of the gap to the best prior model (BrainHarmonix-F: 94.4% on HCP-YA, 23.8% on NSD; Figure 8).

**Spatial patch size.** Table 4d showed improved performance for smaller temporal patch size, suggesting a speed/accuracy tradeoff. In Table 10, however, it appears that this does not extend to smaller *spatial* patch sizes. Note that for spatial patch size 8, we use $2\times$ patch masking (Yang et al., 2025), to maintain the $16 \times 16$ mask units.

**Decoder depth, drop path, and probe depth.** In Table 11, we analyze the effects of different depth-related interventions. In contrast to (He et al., 2022; Feichtenhofer et al., 2022), we do not see any clear differences between different decoder depths (Table 11a). In contrast to (Ryali et al., 2023), we also don't see an effect of increasing drop path

| frames | $p_t$ | Age | Sex | Task21 | COCO24 |
|--------|-------|------|------|--------|--------|
| 16 | 4 | 44.1 | 70.8 | 97.3 | 27.5 |
| 16 | 16 | 43.0 | 71.3 | 97.4 | 27.3 |
| 64 | 16 | 43.8 | 72.9 | 95.7 | 25.1 |

*Table 9.* Worse state decoding for longer input sequences. *frames* refers to number of temporal frames per sample. $p_t$ is the temporal patch size. The model is a parcellation based CortexMAE with the Schaefer-400 parcellation. red indicates $> 3\sigma$ below baseline .

| $p$ | $p_t$ | Age | Sex | Task21 | COCO24 |
|-----|-------|------|------|--------|--------|
| 16 | 16 | 46.7 | 84.9 | 97.9 | 27.6 |
| 16 | 4 | 49.5 | 88.1 | 99.0 | 29.6 |
| 8 | 4 | 42.6 | 84.6 | 98.9 | 29.4 |

*Table 10.* No benefit of smaller spatial patch size $p$ on state decoding, and worse performance on trait prediction.

| depth | acc | | dpr | acc | | depth | acc |
|-------|------|--|-----|------|--|-------|------|
| 2 | 31.6 | | 0.0 | 31.3 | | 0 | 15.2 |
| 4 | 30.1 | | 0.1 | 31.5 | | 2 | 21.2 |
| 8 | 28.8 | | 0.2 | 30.7 | | 4 | 25.9 |
| 12 | 31.2 | | 0.3 | 30.2 | | 8 | 29.5 |
| | | | | | | 12 | 30.6 |

*(a)* Decoder depth.   *(b)* Drop path rate.   *(c)* Probe depth.

*Table 11.* Depth related ablations on NSD COCO24. (a) No clear differences between different decoder depths. (b) No clear effect of increasing drop path rate. (c) Probing deeper encoder layers performs better. Depth 0 corresponds to probing immediately after the ViT patch + position embedding.

rate (resulting in stochastic encoder depth) (Table 11b).

We do, however, observe improved performance from probing the encoder at deeper layers (cf Bolya et al. 2025). In particular, attaching the attentive probe at depth 0 (after the patch + position embedding but before any ViT blocks), results in a ∼50% relative performance drop from baseline. This validates that the encoder learns to compute better representations than are directly accessible in the input data.

**Pretraining data mixture.** The HCP-YA pretraining dataset includes both resting-state and task-based fMRI runs. A natural question is whether one scanning condition produces more useful data for training foundation models than the other. Although resting-state is more common and arguably easier to collect, there is some evidence that data collected during structured cognitive tasks or naturalistic viewing are more predictive of behavior (Greene et al., 2018; Finn & Bandettini, 2021).

Table 12 shows that pretraining on all data results in more predictive representations than either modality separately. Pretraining on task transfers better to the in-distribution HCP-YA Task21 benchmark (which covers the same set of cognitive tasks), while pretraining on rest appears to lead to better generalization on the out-of-distribution NSD

| subset | hours | Age | Sex | Task21 | COCO24 |
|--------|-------|------|------|--------|--------|
| all | 2058 | 47.5 | 87.4 | 98.9 | 31.0 |
| rest | 1072 | 47.7 | 85.5 | 97.5 | 30.8 |
| task | 987 | 47.4 | 85.4 | 98.5 | 28.7 |

*Table 12.* Pretraining on all HCP-YA data outperforms resting-state or task-based only pretraining. The model is a flat map CortexMAE. Baseline performance is from Table 2.

COCO24 visual decoding benchmark. All three pretraining datasets perform similarly on trait prediction benchmarks.

### C.2. Large scale pretraining on UKBB

We use HCP-YA (Van Essen et al., 2013) as our primary pretraining dataset because it is openly accessible to all researchers, supporting easier reproducibility and direct comparison. Some prior works train on datasets (UKBB Miller et al. 2016, ABCD Casey et al. 2018) that are publicly available in theory, but difficult to access in practice. Accessing UKBB imaging data, for example, costs £9,000 per institution[3]. Datasets like UKBB and ABCD are, of course, extremely valuable data resources for the field, and their fees and usage restrictions are well justified. Nonetheless, the barrier to access these data limits machine learning modeling progress, which relies crucially on common open datasets (Deng et al., 2009).

To evaluate the impact of pretraining dataset selection, we pretrained flat map CortexMAE models on varying subsets of HCP-YA and UKBB (Figure 11). Surprisingly, we observe that UKBB pretraining *underperforms* HCP-YA pretraining, even with $10\times$ more data. This holds both for the HCP-YA Task21 benchmark (in-distribution for the HCP-YA models), and NSD COCO24 (OOD with respect to both HCP-YA and UKBB). Importantly, this is a preliminary analysis, and the UKBB pretraining pipeline is much less tested compared to HCP-YA. Nonetheless, the results show that open datasets like HCP-YA are good options for fMRI foundation model pretraining.

Figure 11 also shows that there is currently no clear benefit of longer pretraining on larger data subsets. For the HCP-YA models, continued pretraining on more data yields better performance on in-distribution HCP-YA Task21, but not out-of-distribution NSD COCO24. For the UKBB models, the smallest data subset is competitive on HCP-YA Task21, while outperforming all other models on NSD COCO24. This reinforces the weak generalization of data scaling trends to OOD settings from Section 5.2. Notably, the result in Figure 11, bottom left, is somewhat in tension with Figure 7c. A possible explanation could be that models in Figure 11 are trained for the full cosine lr schedule, whereas

---

[3]https://www.ukbiobank.ac.uk/use-our-data/fees/

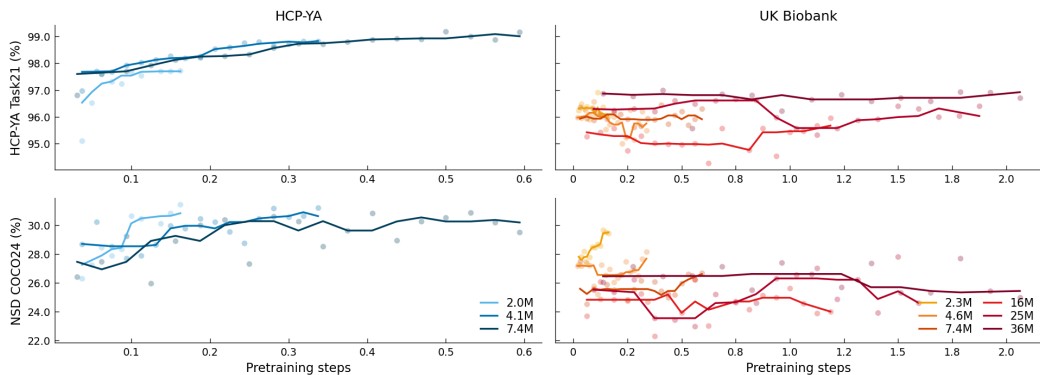

*Figure 11.* Comparing pretraining datasets across data scales. Downstream performance as a function of pretraining steps (in millions) for models trained on subsets of HCP-YA (left) and UKBB (right). Dots are individual checkpoints; lines show a rolling median. All models within each family are trained for a matched number of effective epochs, with total steps scaled proportionally to dataset size.

| decoder | attn | lin |
|---|---|---|
| self | 31.0 | 11.2 |
| cross | 29.8 | 8.6 |
| cross-reg (16) | 25.8 | 10.9 |
| cross-reg (4) | 31.0 | 12.9 |
| cross-reg (1) | 29.1 | 24.2 |

*(a)* Decoder architecture

| decoder | w/o mask | w/ mask |
|---|---|---|
| self | 31.0 | 32.0 |
| cross | 29.8 | 31.3 |
| cross-reg (1) | 29.1 | 28.9 |

*(b)* Decoder edge masking

*Table 13.* Decoder comparison. (a) All decoding approaches perform well with attentive probe. *Only* single-register cross-register decoding supports linear probe. (b) Edge masking helps self and cross attention decoding. Number of registers in parentheses.

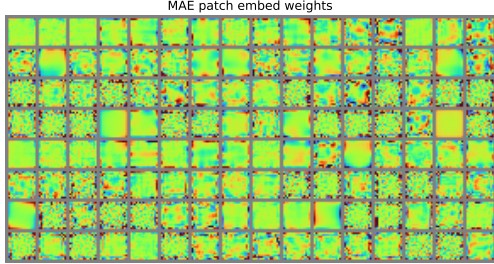

*Figure 13.* Patch embedding $16 \times 16$ filters for the official MAE (He et al., 2022) showing edge artifacts. Maps show the red channel only for consistency with Figure 12.

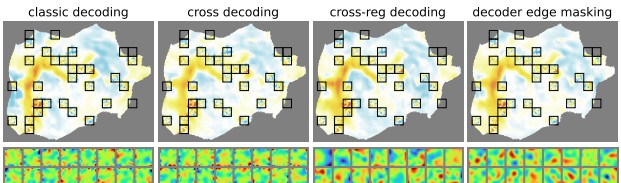

*Figure 12.* Cross-register decoding and decoder edge masking prevent local interpolation and remove edge artifacts in weights. (top) Example reconstructions. Boxes indicate visible patches. (bottom) Example maps from the decoder head weight matrix.

Figure 7c uses intermediate checkpoints without the benefit of a full cooldown.

### C.3. MAE decoding experiments

In this section, we discuss some tangential explorations into MAE decoder architecture and loss masking.

**Cross-register decoding.** We experiment with three variants for the MAE decoder. The standard MAE decoder reconstructs masked pixel values by self-attending over a sequence of embeddings and `[MASK]` tokens. Alternatively, CrossMAE reconstructs by cross-attention only (Fu et al., 2025). This removes interactions between masked patches, reducing compute cost and simplifying the decod-

ing pipeline. We propose a natural third extension of these approaches, which we call cross-*register* decoding. We prepend a set of register tokens to the input (Darcet et al., 2024), and decode by cross-attending *only* over the encoded registers. We hypothesize that compressing the latent information into a small set of registers will promote learning a more discriminative global embedding.

Table 13a compares the three decoder architectures on NSD COCO24. All decoding strategies perform sensibly with the default attentive probe. (The poor performance of cross-reg (16) may stem from underused patch embeddings going into the probe.) However, only the single-register cross-register decoding reaches competitive performance using the weaker linear probe. The absence of an explicit global image embedding in MAE is a known limitation for linear probe (He et al. (2022), Fig. 9). Compressing the encoded information into a single register ("cross-reg (1)") helps address this issue.[4]

**Decoder edge masking.** To prevent the model from interpolating at the edges of observed patches, we propose

---

[4]He et al. (2022) note that linear probe requires a higher masking ratio than fine-tuning. Higher masking means a harder objective *and* more latent compression.

| model | ABIDE Dx | ADHD200 Dx | ADNI Dx | PPMI Dx | HCP-A Age | HCP-A Sex | HCP-YA Task21 | NSD COCO24 |
|---|---|---|---|---|---|---|---|---|
| BrainLM | 48.5 | 60.2 | 55.5 | 52.7 | 43.8 | 64.9 | 92.5 | 22.3 |
| Brain-JEPA | 52.6 | 59.1 | 62.1 | 56.8 | 28.0 | 49.3 | 73.6 | 9.7 |
| BrainHarmonix-F | 50.4 | 59.9 | 51.6 | 56.3 | 37.0 | 58.4 | 94.4 | 23.8 |
| Brain-Semantoks | 57.3 $\pm 0.6$ | 59.7 $\pm 0.7$ | 57.2 $\pm 2.0$ | 55.1 $\pm 1.4$ | 45.7 $\pm 1.3$ | 79.2 $\pm 1.7$ | 89.8 $\pm 0.2$ | 15.3 $\pm 0.5$ |
| CortexMAE-P | **62.0** $\pm 0.8$ | 56.8 $\pm 0.6$ | 61.6 $\pm 1.2$ | **61.4** $\pm 1.3$ | 44.2 $\pm 0.5$ | 71.2 $\pm 1.0$ | **97.5** $\pm 0.2$ | **27.5** $\pm 0.5$ |
| SwiFT | 53.7 | 59.2 | 60.7 | 55.8 | 38.1 | 74.5 | 63.6 | 7.7 |
| NeuroSTORM | 53.5 | **61.1** | **66.8** | 53.4 | **56.6** | 82.7 | 70.5 | 12.2 |
| CortexMAE-V | 60.4 $\pm 0.8$ | 58.8 $\pm 1.1$ | 64.3 $\pm 1.6$ | 59.1 $\pm 1.2$ | 53.4 $\pm 0.5$ | 86.3 $\pm 0.7$ | 96.2 $\pm 0.3$ | 27.7 $\pm 0.7$ |
| CortexMAE-F | 61.4 $\pm 1.3$ | 59.2 $\pm 1.0$ | 62.4 $\pm 1.4$ | 58.8 $\pm 1.1$ | 47.5 $\pm 1.6$ | **87.4** $\pm 0.7$ | **98.9** $\pm 0.1$ | **31.0** $\pm 0.7$ |
| connectome | 59.8 | 57.0 | 58.6 | 58.0 | 45.6 | 81.9 | 82.4 | 7.4 |

*Table 14.* fMRI foundation model comparison using BRAINMARKS. (top) parcellation based models, (middle) dense volume models plus our flat map CortexMAE-F, (bottom) functional connectome baseline. The best models in each category are indicated with **bold** and underline. Only scores $> 3\sigma$ better than connectome are highlighted. Results are the same as reported in Figure 8.

| model | coord | ABIDE Dx | ADHD200 Dx | ADNI Dx | PPMI Dx | HCP-A Age | HCP-A Sex | HCP-YA Task21 | NSD COCO24 |
|---|---|---|---|---|---|---|---|---|---|
| SwiFT | ✗ | 53.7 | 59.2 | 60.7 | 55.8 | 38.1 | 74.5 | 20.7 | 6.4 |
| Brain-JEPA | ✗ | 52.6 | 59.1 | 62.1 | 56.8 | 28.0 | 49.3 | 16.9 | 6.4 |
| NeuroSTORM | ✗ | 53.5 | 61.1 | 66.8 | 53.4 | 56.6 | 82.7 | 17.5 | 6.4 |
| SwiFT | eval | 53.8 | 61.5 | 58.2 | 52.1 | 38.4 | 66.7 | 63.6 | 7.7 |
| Brain-JEPA | eval | 53.5 | 50.4 | 51.0 | 52.6 | 27.6 | 50.2 | 73.6 | 9.7 |
| NeuroSTORM | eval | 53.5 | 57.7 | 53.5 | 53.2 | 52.7 | 73.5 | 70.5 | 12.2 |

*Table 15.* Effect of coordinate normalization on models pretrained without it. (top) performance with models' official preprocessing, without coordinate norm. (bottom) performance with coordinate normalization added.

*decoder edge masking*: we mask out a border of 4 pixels surrounding each observed patch and exclude them from the reconstruction loss. fMRI data are spatially very smooth, particularly after surface-based preprocessing. As a result, local interpolation is a noticeable problem in this setting.

Figure 12 shows that decoder edge masking eliminates interpolation at the edges of observed patches and removes edge artifacts from the decoder head weights for self and cross-attention decoding. In Table 13b, we see this translates into modest improvement for downstream prediction. Interestingly, cross-register decoding does not show the same interpolation artifacts, and edge masking does not appear to impact prediction performance.

Edge artifacts are also visible in the original MAE patch embedding from He et al. (2022) (Figure 13). This suggests that even for natural images, MAEs partly exploit a local interpolation shortcut strategy.

### C.4. Supplementary benchmark analyses

Table 14 shows the same results from Figure 8 in table form.

**Evaluation-time coordinate normalization.** Table 15 tests the effect of applying coordinate normalization during evaluation on models that were pretrained without it. Similar to

| model | params | FLOP | data/s | fwd/s | FLOP/s |
|---|---|---|---|---|---|
| BrainLM | 113M | 382G | 267K | 96K | 73T |
| Brain-JEPA | 87M | 1511G | 338K | 86K | 261T |
| BrainHarmonix-F | 89M | 3135G | 310K | 20K | 122T |
| Brain-Semantoks | 63M | 6G | 179K | 1079K | 12.2T |
| CortexMAE-P | 85M | 8062G | 380K | 19K | 303T |
| SwiFT | 4M | 183G | 0.5K | 15K | 5.5T |
| NeuroSTORM | 8M | 141G | 0.4K | 24K | 6.7T |
| CortexMAE-V | 87M | 9892G | 1.5K | 15K | 292T |
| CortexMAE-F | 86M | 7234G | 3.8K | 20K | 292T |

*Table 16.* Compute comparison of fMRI foundation models. FLOP count is for forward pass on a single sample (500 frames). Data loading and forward pass throughput is in frames per second. FLOP/s compute throughput is for model forward pass. Models are evaluated with AMP enabled at bfloat16. CortexMAE-{P,F,V} refer to the parcel, flat map, and volume models respectively.

Table 6, the models' performance on state prediction is near chance using the models' official preprocessing pipeline without coordinate norm, and substantially improves with coordinate norm enabled. Interestingly, however, coordinate norm hurts performance on the trait prediction tasks, suggesting that the models are partly relying on static structural features.

*Figure 14.* Attentive probe performance on subject-level trait prediction. Markers indicate each model's performance (balanced accuracy) on a fixed train/test split using the attentive probe protocol. Confidence intervals indicate the mean $\pm 2 \times$ stdev for the logistic probe over the 100 randomized splits. Attentive probe performance is variable, but within logistic probe CIs.

**Model compute comparison.** Table 16 compares all fMRI foundation models in terms of their compute performance. Parcellation based models (top) use ViT-B-scale parameter counts, and have efficient data loading due to the sparse representation. Prior volume models (middle) have low parameter counts ($<$10M) and are bottlenecked by data loading. CortexMAE models (bottom) maintain the same model ViT-B architecture across input representations, achieve a reasonable compute/data tradeoff, and have better GPU utilization ($>$290TFLOP/s).

**Attentive probe performance on trait prediction.** Our main evaluations for trait prediction use a simple linear logistic probe with randomized train/test splits for better robustness to small sample sizes (Table 1). Figure 14 evaluates all models on trait prediction instead using the attentive probe protocol. Attentive probe performance is variable across models, but within the confidence intervals of the logistic probe. This result further highlights the unreliability of the current clinical diagnosis prediction benchmarks.

# D. Compute cost

One pretraining run for our default flat map CortexMAE currently takes $\sim$28 hours on our system using 1 NVIDIA H100 GPU (10GB memory usage, $\sim$160ms / step). This time varies depending on system load, since pretraining is IO bound. Total compute used for all experiments was $\sim$7K H100 hours.

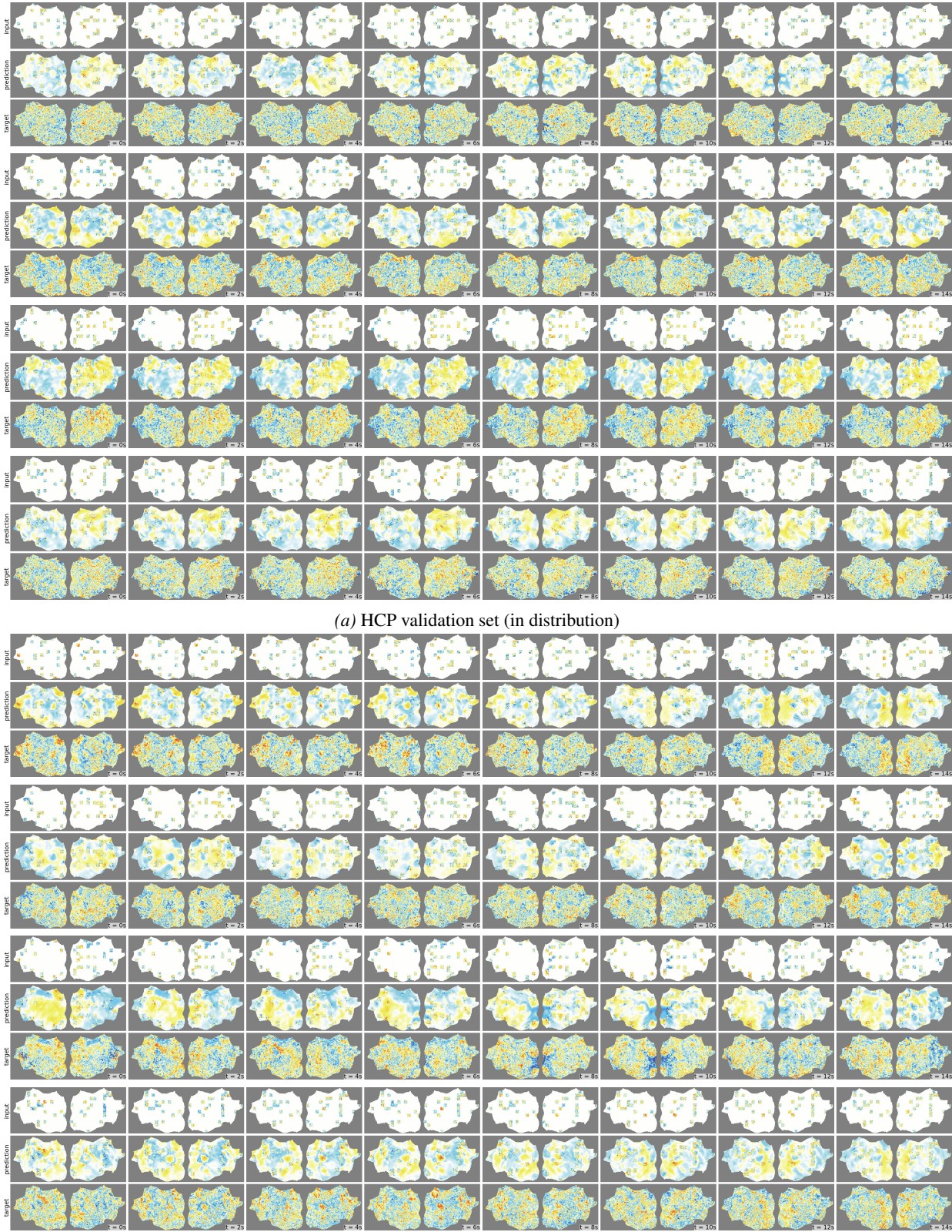

*(a)* HCP validation set (in distribution)

*(b)* NSD validation set (out-of-distribution)

*Figure 15.* Uncurated examples of MAE predictions on fMRI flat maps.

