# OpenReview forum: "Scaling Vision Transformers for Functional MRI with Flat Maps"
_ICML.cc/2026/Conference — ICML 2026 regular_

### Official Review · Reviewer_PMPX · 2026-02-22

**Soundness:** 3
**Presentation:** 2
**Significance:** 3
**Originality:** 3
**Overall Recommendation:** 4
**Confidence:** 4

**Summary:**

The paper introduces CortexMAE, a foundation model designed to analyze brain activity by projecting 3D fMRI data into 2D cortical flat maps. This approach allows the use of standard Vision Transformers and masked autoencoders on fMRI. And it provides reproducible benchmarks and baselines. Although the model did not achieve state-of-the-art under certain conditions, its exploration of scaling and benchmarks still has some value.

**Compliance With Llm Reviewing Policy:**

Affirmed.

**Final Justification:**

Soundness: This is a solid study, but the use of Pycortex images carries risks of distortion and approximation, which may be viewed skeptically by traditional neuroscientists. I am not an expert about this point. It is unclear whether the proposed approach for handling inputs with a 4-second temporal resolution is appropriate.

Presentation: The overall presentation is weak and requires significant refinement. Specifically, the description and experiment settings of the baseline methods are insufficient, and the figures appear unpolished, lacking the clarity and professional finish expected for this venue.

Significance: Problem is important.

Originality: First paper for foundation model by Pycortex images. No significant algorithmic innovation.

According to the rebuttal, I increased my score to weak accept and major revisions are still necessary for the final version. And the reply rebuttal comment introduced new concern about handling 4-second inputs.

**Key Questions For Authors:**

1. I encourage the authors to include results from full fine-tuning to establish the model's true performance ceiling. Furthermore, the evaluation would be significantly strengthened by incorporating a broader range of parcel-free foundation models.
2. The paper notes that long-sequence models were adapted to 16s clips via padding for state prediction tasks. Could the authors clarify whether this adaptation introduces distributional shift that disadvantages these baselines.
3. In the NSD dataset, each stimulus is presented with a gap of approximately 4 seconds. However, the paper utilizes 16-second fMRI clips. How is this temporal alignment achieved?
4. There appears to be a significant discrepancy between the resting pre-training phase and the state prediction task. While Table 9 compares other model trained from scratch, could you provide results for the same model (ours) without pre-training? This would allow for a direct assessment of the performance gains specifically attributed to the pre-training stage
5. Table 4 states that patch norm is better, but why isn't this loss function used in the main experiment?

**Limitations:**

yes

**Strengths And Weaknesses:**

**Strengths**

1. The problem is important, and the experiments are comprehensive.
2. The study performs a quantitative scaling analysis for fMRI, observing a strict power law scaling for reconstruction loss.
3. The model shows strong performance on state prediction tasks.

**Weaknesses**

**Major**
1. The results are only evaluated using probing rather than full fine-tuning, which leaves the performance ceiling unknown.
2. The comparison of state detection seems unfair as the baselines are trained on long-sequence data. Additionally, the results for disease diagnosis under attentive probing do not achieve SOTA.
3. While the paper performs an ablation study on the influence of parcel, flat, and volume representations, it compares against a voxel model that is not a foundation model. There are several recent parcel-free models, though mentioned, they are not included in the comparison. Since this paper introduces a benchmark, including some voxel-level models would be helpful.

**Minor**
1. Using images as input for fMRI is not a new idea [1].
2. Missing descriptions for some of the baseline models.
3. The related work section is limited, which is unfriendly for beginners. I suggest conducting a more thorough survey, especially as the paper introduces a benchmark suite.

[1] MinD-3D++: Advancing fMRI-Based 3D Reconstruction with High-Quality Textured Mesh Generation and a Comprehensive Dataset

---

> ### Author Rebuttal · Authors · 2026-03-30
>
> Thank you for the helpful feedback. We appreciate the positive comments on our comprehensive experiments and strict power law scaling results.
>
> > The results are only evaluated using probing rather than full fine-tuning
>
> We have now performed a preliminary test of fine-tuning (rank-8 LORA on QKV layers), and found only a small performance increase over our attentive probe (32.9% vs 30.7% on NSD COCO24). The small gap can be partly attributed to our use of attentive probe [1], rather than classic linear probe which performs much worse (11.8%, Table 5a).
>
> More generally, however, we emphasize that our goal is to do a systematic model comparison, not just obtain absolute ceiling performance. Under fine-tuning, it is difficult to disentangle the impact of foundation model pretraining vs general model capacity or fine-tuning compute budget. By contrast, probes offer a controlled method to specifically assess pretrained representation quality [2]. Moreover, probes are cheap to evaluate, making it practical to do dense hyperparameter tuning for all models.  This makes probing more appropriate for SSL model comparison in our view.
>
> > The comparison of state detection seems unfair as the baselines are trained on long-sequence data.
>
> To investigate the reviewer’s question, we trained parcel MAEs on longer sequence inputs (64 TRs = 64s) with temporal patch size of 16 TRs. We find that longer sequence training does indeed result in performance loss on trial-level state prediction tasks. However, the difference is only ~2% and performance is still robustly above the simple connectome and MLP baselines. By contrast, the prior models are at chance on these tasks.
>
> | frames |  p_t | HCP-A Age | HCP-A Sex | HCP-YA Task21 | NSD COCO24   |
> |---:|---:|:---|:---|:---|:---|
> | 16 | 4 | 44.1 ± 0.7  | 70.8 ± 0.6  | 97.3 ± 0.1 | 27.5 ± 0.2   |
> | 64 | 16 | 43.8 ± 0.1  | 72.9 ± 0.8  | 95.7 ± 0.2 | 25.1 ± 0.6   |
>
> We agree, however, that there is a need to explain prior models’ performance on these tasks. In our response to Reviewer g5t6, we report a new ablation showing that data normalization is the most likely explanation.
>
> > the results for disease diagnosis under attentive probing do not achieve SOTA.
>
> The lack of open, standardized clinical diagnosis benchmarks makes it impossible to determine an unambiguous SOTA. Although many prior works use the same source datasets, each work has its own benchmark preparation. Further, “achieving SOTA” misses the conclusion of our paper: none of the existing methods are able to clearly differentiate themselves from a simple connectome baseline.
>
> To our knowledge, our work is the first fMRI FM effort to release an open, fully reproducible evaluation suite. We hope that our effort helps to advance the community’s adoption of reproducible benchmarking [3].
>
> >  There are several recent parcel-free models, though mentioned, they are not included in the comparison.
>
> To our knowledge, NeuroSTORM [4] is the only published parcel-free fMRI FM missing from our evaluation. At the time of submission, we were unable to include it due to computational cost, but we have since added it for the final version.
>
> | model | HCP-A Age | HCP-A Sex | HCP-YA Task21 | NSD COCO24   |
> |:---|:---|:---|:---|:---|
> | NeuroSTORM  | 56.6 | 82.7 | 70.5 | 12.2  |
> | CortexMAE-V | 53.5 ± 1.2  | 85.9 ± 0.4  | 95.9 ± 0.2 | 26.5 ± 0.8   |
> | CortexMAE   | 44.1 ± 1.2  | 86.2 ± 1.0  | 98.6 ± 0.1 | 30.0 ± 1.0   |
>
> Our model is largely on par with NeuroSTORM on subject-level trait prediction. The one exception is HCP-A age, where our volume MAE (CortexMAE-V) is on par. On state prediction, our models significantly outperform NeuroSTORM.
>
> Finally, we are aware of the promising concurrent work Omni-fMRI [5]. We intend to include it in our evaluation for the final version.
>
> > How is this temporal alignment achieved?
>
> The 16s trial window is synchronized to the start of the trial. As a result, the peak of the HRF is well included. No other temporal alignment is applied.
>
> > could you provide results for the same model (ours) without pre-training
>
> We agree that the trained-from-scratch performance of our flat map ViT is a useful reference point. We have not run this experiment, but we will include the result in the final version.
>
> > Table 4 states that patch norm is better, but why isn't this loss function used in the main experiment?
>
> Since none of the ablations in Section 3 produced exceptionally clear improvements, we decided to focus on the simple default model for all other experiments. We will clarify this rationale in the final version.
>
> >  MinD-3D++: Advancing fMRI-Based 3D Reconstruction
>
> We thank the reviewer for bringing this reference to our attention. We plan to expand our related work section as the reviewer also suggests, and will include this reference.
>
> [1] Darcet et al. TMLR 2025.
>
> [2] Balestriero et al. arXiv 2023.
>
> [3] Pineau et al.  JMLR 2021.
>
> [4] Wang, C et al. arXiv 2025.
>
> [5] Wang, M et al. arXiv 2026.

---

> > ### Author Rebuttal · Reviewer_PMPX · 2026-04-01
> >
> > Thank the authors for the rebuttal.
> >
> > 1. The discussion of normalization is helpful, and your conclusion appears broadly consistent with [1]. However, in my own experiments, this pattern does not always hold, especially for certain time-series models. A more complete analysis of normalization strategies, including voxel-wise, frame-wise, and global normalization, would be valuable. It would strengthen the final version if these comparisons could be included, although I view this as optional rather than essential.
> >
> > 2. I am still confused about the temporal alignment. How does the model handle inputs with a 4 s temporal resolution? will the sequence include 4 stimulations? For example, are they padded to 16 s, repeated, or processed in some other way?
> >
> > I am inclined to increase my score. I would still appreciate seeing the results of the promised experiments by the end of the discussion period, it could be only the results for HCP Sex. But even without them, the rebuttal has addressed a substantial part of my concerns.
> >
> > [1] Self-Supervised Transformers for fMRI representation 2022

---

> > > ### Author Response · Authors · 2026-04-02
> > >
> > > We’re glad that the rebuttal was useful, and we appreciate your openness to our paper. Happy to follow-up on these points.
> > >
> > > > A more complete analysis of normalization strategies, including voxel-wise, frame-wise, and global normalization
> > >
> > > Yes, the normalization analysis deserves to be fleshed out more. We had the results you ask for on-hand, but couldn’t include in our initial response due to the space constraint.
> > >
> > > Below is a more complete ablation table comparing different normalization strategies on our default flat map CortexMAE. We see that coordinate normalization is essential for state prediction. Frame-wise normalization provides a smaller but still significant improvement over global normalization. On trait prediction, our model benefits from coordinate normalization, but performance is still well above chance without it.
> > >
> > > | coord   | frame   | global   |   HCP-A Age |   HCP-A Sex |   HCP-YA Task21 |   NSD COCO24 |
> > > |:---|:---|:----|-------:|-------:|---:|---:|
> > > | ✔  | ✔  | ✘   |   48.1 |   87.6 |   98.8 |  31.4 |
> > > | ✔  | ✘  | ✔   |   50.8 |   87.3 |   97.7 |  26.3 |
> > > | ✘  | ✔  | ✘   |   38.4 |   77.5 |   15.5 |   5.4 |
> > > | ✘  | ✘  | ✔   |   40.2 |   79.1 |   16.1 |   5.5 |
> > >
> > > The interpretation is that omitting coordinate normalization (i.e. voxel-wise normalization) effectively blinds the model to dynamic BOLD signal changes needed for state prediction, which are only 1-2% compared to T2*-weighted tissue contrast. Indeed, this is the same rationale as [1]; we will reference this work on these points in our final version.
> > >
> > > > How does the model handle inputs with a 4 s temporal resolution?
> > >
> > > Sorry for not explaining more clearly in the first response. We extract the full 16s fMRI clip following target stimulus onset, without any cropping or padding. Due to the slow HRF, the observed fMRI activity therefore reflects a mixture of overlapping responses to multiple stimuli, which are presented every 4s. The models must learn to “attend” only to the activity evoked by the target stimulus presented at window onset, and ignore the other responses.
> > >
> > > > I would still appreciate seeing the results of the promised experiments by the end of the discussion period
> > >
> > > We can’t promise, but we will make an effort to share results for ViTs trained from scratch before the end of the discussion period.

---

### Official Review · Reviewer_os1b · 2026-03-08

**Soundness:** 3
**Presentation:** 2
**Significance:** 3
**Originality:** 2
**Overall Recommendation:** 4
**Confidence:** 2

**Summary:**

To address the challenge of training foundation models for functional magnetic resonance imaging (fMRI), this paper proposes an intermediate tokenization strategy that converts 3D fMRI volume data into 2D cortical flat maps, upon which the CortexMAE model is constructed. The model is pretrained on 2.3 thousand hours of fMRI flat map videos using a spatiotemporal masked autoencoder (MAE). The authors compare the performance of CortexMAE with that of identically structured MAE models trained on parcel-averaged and native volume data, and conduct the first quantitative scaling analysis in the fMRI domain, revealing a strict power-law scaling relationship. Meanwhile, they develop the first open evaluation suite for fMRI foundation models, and perform a comprehensive comparison of cognitive state decoding and clinical trait prediction across seven datasets covering different imaging sites and scan parameters. The study finds that CortexMAE significantly outperforms existing models on cognitive state decoding tasks, yet all models exhibit inconsistent performance in clinical trait prediction. Additionally, the authors propose two improvements to the MAE framework, namely cross-register decoding and decoder edge masking, and open-source the relevant code and benchmark datasets, thereby establishing reproducible benchmarks and a strong baseline model for the research of fMRI foundation models.

**Compliance With Llm Reviewing Policy:**

Affirmed.

**Final Justification:**

The authors’ rebuttal and follow-up clarifications address most of my main concerns, particularly regarding the intended scope of the paper, the interpretation of the scaling results, and which open questions are beyond the scope of the current work. As a result, I find the paper’s contribution and positioning clearer, and I am raising my score to 4.

At the same time, I am not particularly familiar with all of the tasks and subareas covered in the paper, so I keep my confidence at 2.

**Key Questions For Authors:**

See Weaknesses

**Limitations:**

yes

**Strengths And Weaknesses:**

Strengths:
1. Rationality: The research design is logically robust. The proposed planar mapping tokenization strategy effectively balances the information richness of raw fMRI data with the computational efficiency of voxel-averaging methods. The pre-training and evaluation setups are standard, and the scaling analysis and ablation studies are well-designed to validate the model and strategy. The experimental results largely align with the analysis, with no significant technical flaws.
2. Presentation: The paper is clearly structured, progressing logically from problem statement and method design to experimental implementation and analysis. Figures and tables effectively present the experimental results, with detailed descriptions of the model architecture and experimental settings. The open-sourced code and dataset enhance reproducibility. However, the analysis of some results is somewhat superficial; for instance, the reason for the volumetric model's superior performance on HCP-A age prediction is only speculated upon without in-depth validation.
3. Significance & Originality: This work presents an open evaluation suite for fMRI foundation models, filling a gap in standardized benchmarks for the field. The scaling analysis provides crucial insights for large-scale training of fMRI models. The proposed CortexMAE achieves SOTA performance in cognitive state decoding and reveals common issues in current models for clinical prediction, guiding future research. The core originality lies in adapting Vision Transformer patch embeddings for fMRI planar representations, creating an efficient tokenization strategy. The cross-registration decoding and decoder edge masking are lightweight yet innovative extensions of the MAE/ViT framework for fMRI data. While built on established architectures, the tailored adaptations for fMRI, combined with the first scaling analysis and open benchmark, represent significant original contributions.

Weaknesses:
1. Insufficient Depth in Clinical Feature Prediction: The study merely reports that all models, including CortexMAE, perform poorly on clinical or disease prediction tasks—often underperforming simple connectome baselines—without any targeted investigation. It does not examine data-related factors such as small sample size, high heterogeneity, or low signal-to-noise ratios in clinical fMRI; nor does it explore model-level issues, such as a potential mismatch between representations learned for dynamic cognitive states and those needed for static clinical traits. The suitability of linear probing for small-sample clinical tasks is also left unexamined. The analysis stops at observation, with no effort to trace or validate the underlying causes.
2. Superficial Ablation Analysis: Although ablation studies are conducted on masking strategies, data augmentation, reconstruction targets, and decoder designs, the discussion remains at the level of numerical results and surface-level conclusions. Key phenomena are left unexplained: why standard augmentations harm performance (vaguely attributed to "orthogonality to fMRI manifolds"); why the volume model outperforms on HCP-A age prediction (only speculated without controlled validation); why decoder edge masking and cross-registration yield only marginal gains. The ablations confirm effectiveness but offer little insight into model behavior or guidance for future design.
3. Lack of Clinical Translation Experiments: All experiments are conducted on public, standardized fMRI datasets. No validation is performed in real-world clinical settings, ignoring challenges such as cross-site scanner variability, differences in acquisition protocols, patient heterogeneity, non-standard preprocessing, and the presence of noise or artifacts. Practical considerations—like inference speed, interpretability, or integration into clinical workflows—are not addressed, leaving the model's real-world applicability unvalidated.
4. One-Sided Interpretation of Model Comparisons: While the evaluation setup is unified for fairness, the interpretation overlooks the design principles and intended use cases of the compared models. BrainLM and Brain-JEPA, designed for long-sequence fMRI, are evaluated on 16-second tasks, and their poor performance is presented as a model deficiency rather than a task–model mismatch. SwiFT, designed for short sequences, is included, but its underperformance is not analyzed—only numerical differences are noted. This limits the objectivity and depth of the comparison.
5. Scaling Analysis Not Operationalized: Although power-law scaling is observed and model performance saturates at 43M parameters, the findings are not translated into actionable guidance. No investigation is conducted into whether the scaling law generalizes across tasks (e.g., decoding vs. clinical prediction) or datasets. No recommendations are provided on optimal data size or model capacity for different scenarios, leaving the scaling results as theoretical observations without practical utility for model development.

---

> ### Author Rebuttal · Authors · 2026-03-30
>
> Thank you for the helpful feedback. We appreciate the positive comments on the rationality of our flat map approach, clear presentation, scaling analysis, and open benchmark suite.
>
> > Insufficient Depth in Clinical Feature Prediction
>
> We agree that analyzing the inconsistent performance of all fMRI foundation models on clinical diagnosis prediction is a crucial next step. As discussed in our response to Reviewer g5t6, we agree that small sample size, high variability, and weak brain-trait association are likely explanations. The current work does not have the scope to interrogate these issues fully, however. By making our benchmark suite open, we hope to encourage the community to scrutinize these results.
>
> > Superficial Ablation Analysis
>
> We respectfully disagree with the reviewer’s description of the ablations as “superficial”. All other reviewers describe the experiments as “comprehensive”. We acknowledge, however, that the ablation experiments raise unresolved questions (which we note explicitly in the text). This is common for ablation experiments, and in many ways a sign of their utility. For example, the ablations in the original MAE work [1] show a benefit of decoder depth, a benefit of patch normalization, and a discrepancy between fine-tuning and linear probe performance. These results are presented without full explanation, but have since been elaborated on by other studies.
>
> > Lack of Clinical Translation Experiments
>
> We agree that clinical translation is the ultimate end goal of this line of research. However, the goal of the current study is to contribute to the basic science of fMRI foundation models. As our results show, there are still basic questions to resolve before the technology is ready for real world clinical application. For example, establishing robust improvements over baselines in standardized academic benchmarks.
> .
> > One-Sided Interpretation of Model Comparisons
>
> The difference in input sequence length between prior models and ours is a good point. Since Reviewer PMPX raises the same question, we provide our full response there.
>
> > Scaling Analysis Not Operationalized
>
> We appreciate the reviewer’s feedback, and will revise our description of the scaling results to be more clear and actionable. To summarize our main takeaways:
>
> - in-distribution test set MAE reconstruction scales strongly with pretraining dataset size, but this does not generalize to OOD data. Action: we must explore more diverse pretraining datasets.
> - Performance scales with model size but only up to ~43M parameters. Action: relatively small models are sufficient for our current data regime.
>
> [2] He et al. CVPR  2022.

---

> > ### Author Rebuttal · Reviewer_os1b · 2026-04-01
> >
> > Thank you for the detailed rebuttal. I appreciate the clarifications, especially regarding the intended scope of the paper, the added discussion of the scaling results, and the explanation of why some of the open questions are beyond the scope of the current work. These responses address my main concerns sufficiently, and I am comfortable raising my score.

---

### Official Review · Reviewer_g5t6 · 2026-03-08

**Soundness:** 3
**Presentation:** 3
**Significance:** 4
**Originality:** 3
**Overall Recommendation:** 4
**Confidence:** 3

**Summary:**

The paper introduces CortexMAE, a framework that adapts ViT to fMRI data by leveraging a 2D representation. The authors propose a Cortical Flat Map Projection that transforms 3D fMRI volumes into 2D images of brain activity. This allows the model to utilize standard ViT architectures and Spatiotemporal Masked Autoencoders (MAE) for self-supervised pre-training.

**Compliance With Llm Reviewing Policy:**

Affirmed.

**Final Justification:**

My concerns have been addressed, so I keep my positive score.

**Key Questions For Authors:**

See the weakness section.

**Limitations:**

Although this study has verified the power-law scaling effect in the fMRI field through large-scale engineering practice, it lacks innovation in methodology and mainly relies on the simple integration of existing technologies.

**Strengths And Weaknesses:**

Strengths:
The flat map projection provides an efficient "middle ground" between low-resolution parcel data and high-resolution 3D volumes. It preserves cortical topology while enabling the use of mature 2D ViT architectures.
The study is supported by exceptionally comprehensive experiments. The authors evaluate the model across several large-scale datasets, including HCP-YA and NSD. By testing on diverse tasks, from cognitive state decoding to clinical trait prediction, the paper provides a robust assessment of the model's capabilities.

Weaknesses:
1. Does the focus on planar maps rule out the subcortical structures? These structures are crucial for a comprehensive understanding of brain function.
2. This paper found that in the comparison of feature prediction, the performance of all models varied, and even on average, it was lower than the simple connectome and MLP baseline. This is a very interesting phenomenon, but the authors lacked an in-depth analysis of it. Why are models that have undergone extensive data pre-training inferior to simple baselines?
3. While the paper provides an impressively large-scale empirical study and successfully identifies scaling laws in fMRI, the technical novelty appears incremental. Both the cortical flat map projection and the Masked Autoencoder (MAE) framework are well-established methodologies in neuroimaging and computer vision, respectively. Could the authors explicitly clarify the unique methodological challenges addressed in this work? Specifically, beyond the integration of existing tools, what were the non-trivial adaptations required to make CortexMAE effective for fMRI data? Without a clearer distinction of these technical contributions, the work risks being perceived primarily as a large-scale benchmarking exercise rather than a fundamental algorithmic advancement.

---

> ### Author Rebuttal · Authors · 2026-03-30
>
> Thank you for the helpful feedback. We appreciate the positive comments on our efficient middle ground flat map approach, comprehensive experiments, and scaling law demonstration.
>
> > Does the focus on planar maps rule out the subcortical structures?
>
> The current flat map representation does exclude subcortical structures. This is an important limitation that we will add to our discussion in the final version.
>
> To test the impact of missing subcortex, we now report newly trained MAEs using parcellations with and without subcortical structures.
> | space | HCP-A Age | HCP-A Sex | HCP-YA Task21 | NSD COCO24 |
> |:---|:----|:---|:---|:---|
> | Schaefer 400 | 44.1 ± 0.7  | 70.8 ± 0.6  | 97.3 ± 0.1 | 27.5 ± 0.2   |
> | Schaefer 400 + Tian S3 | 43.5 ± 0.6  | 72.5 ± 1.7  | 97.5 ± 0.3 | — |
> | A424 | 44.3 ± 0.6  | 71.4 ± 0.3  | 96.7 ± 0.3  | —  |
>
> We see no significant differences in performance between the Schaefer 400 (cortex only), Schaefer 400 + Tian S3 (cortex + subcortex, used by Brain-JEPA), and A424 (cortex + subcortex + cerebellum, used by BrainLM). We believe the lack of performance gain in this test justifies the current focus on cortex-only models.
>
> That said, we appreciate that subcortical structures are key nodes underlying many brain-behavior relationships [1], and that other tasks may be more sensitive to subcortical signals. We hope future work adapts flat maps to subcortical structures.
>
> > Why are models that have undergone extensive data pre-training inferior to simple baselines?
>
> With respect to dynamic state prediction (HCP-YA Task21 and NSD COCO24), we have identified data normalization as a key factor explaining prior models’ poor performance. As we discuss in Section 4, our model uses per-coordinate normalization (z-score each voxel across time), whereas most prior works use only global normalization. We have now performed an explicit ablation of this choice. We find that without coordinate normalization, our model achieves chance performance on state decoding, just like prior models.
>
> | coord norm | HCP-A Age | HCP-A Sex | HCP-YA Task21 | NSD COCO24 |
> |:---|:---|:---|:---|:---|
> | yes | 48.1  | 87.6 | 98.8 | 31.4 |
> | no | 40.2  | 79.1 | 16.1 | 5.5 |
>
> With respect to subject-level trait prediction, our work is the first showing a lack of separation between fMRI FMs and simple baselines. Prior works such as BrainLM, Brain-JEPA, and Brain-Harmony do not include simple baselines. We suspect that the current result reflects a combination of small dataset size, high data variability, and weak brain-trait association.
>
> Even without in-depth analysis, we feel the current “null result” is important and well-supported. Our evaluation is tightly controlled and reproducible: linear probe with reliable logistic regression implementation, dense fair hyperparameter tuning for all models, 100 repeats over random train/test subjects splits, transparent data preparation scripts, open model implementations using official first-party code, all integrated into an open reproducible benchmark suite. We hope our finding encourages the community to coordinate on validated, reproducible benchmarks to address this issue.
>
> > technical novelty appears incremental. Both flat map projection and the MAE framework are well-established
>
> We fully acknowledge that our work does not introduce a technically sophisticated new method. Rather, our method is in fact a simple yet novel combination of two well-established techniques from separated fields: ViTs from computer vision, and flat maps from neuroimaging.
>
> The ICML guidelines state novel contributions include “novel combination of existing techniques”. Many impactful papers can be seen as novel combinations of existing ideas. E.g. ViT adapting transformers to images, MAE adapting BERT to images, CLIP adapting contrastive learning to the image-text setting.
>
> Our method is not only the first fMRI foundation model to use flat maps. It is also the first approach to use an intermediate representation between coarse parcellation and dense volume methods. As further support, we have now pretrained MAEs for each representation across 8 random repeats. The flat map MAE achieves robust improvement over both extremes on dynamic state prediction (p < 0.0001).
>
> | space  | HCP-A Age  | HCP-A Sex  | HCP-YA Task21 | NSD COCO24 |
> |:---|:---|:---|:---|:---|
> | parcel | 43.5 ± 0.8 | 70.8 ± 1.3 | 97.5 ± 0.3 | 27.2 ± 0.6 |
> | volume | 53.5 ± 1.2 | 85.9 ± 0.4 | 95.9 ± 0.2 | 26.5 ± 0.8 |
> | flat   | 44.1 ± 1.2 | 86.2 ± 1.0 | 98.6 ± 0.1 | 30.0 ± 1.0 |
>
> Our work therefore makes a strong argument to the fMRI FM field to further explore such intermediate representations. We hope that the simplicity of our initial approach, combined with the reproducible training and evaluation pipeline, will encourage researchers to build on it.
>
> [1] Maguire E, et al. PNAS 2000.

---

> > ### Author Rebuttal · Reviewer_g5t6 · 2026-04-01
> >
> > My concerns have been adequately addressed. I decided to maintain my score.

---

### Official Review · Reviewer_8Ez3 · 2026-03-12

**Soundness:** 3
**Presentation:** 3
**Significance:** 3
**Originality:** 3
**Overall Recommendation:** 4
**Confidence:** 4

**Summary:**

This study proposes a simple yet underexplored strategy for training foundation models for fMRI data by converting 3D fMRI volumes into 2D representations using a standard cortical flat map projection. Based on this representation, the authors introduce CortexMAE, a spatiotemporal masked autoencoder designed to model fMRI activity over time. The paper conducts a comprehensive empirical comparison across different tokenization strategies for fMRI inputs and evaluates their effectiveness for masked modeling. In addition, the work provides a new benchmark for evaluating scaling behavior and downstream task performance for fMRI FMs.

**Compliance With Llm Reviewing Policy:**

Affirmed.

**Final Justification:**

Thank the authors for the rebuttal. My concerns have been largely addressed. Overall, I find this work to be a meaningful exploration with a valuable benchmark, and I will increase my score accordingly.

**Key Questions For Authors:**

1. In Table 4, the temporal patch size ablation shows that a patch size of 1 (i.e., frame-wise tokens) gives the best performance. I’m curious how the authors interpret this result, and why the final model still uses a patch size of 4. Is this mainly a trade-off between sequence length (efficiency) and performance? Can author possibly discuss in what scenarios using a patch size of 1?

2. The paper proposes a new benchmark for evaluating fMRI foundation models, however, I was wondering how CortexMAE would perform on some downstream tasks commonly used in previous fMRI foundation model papers that are not included here (e.g., HCP-A tasks such as Age, Neuroticism, Flanker, or UKB-based tasks). Could the authors clarify the rationale behind the choice of the current downstream tasks and why some previously used benchmarks were not included?

3. Another question concerns the flat-map representation. Converting 3D fMRI volumes into 2D cortical flat maps removes subcortical regions. Do the authors think that excluding subcortical signals could lead to a potential drop in performance for some tasks? It would be helpful if the authors could discuss the potential impact of this design choice.

**Limitations:**

The paper does not explicitly discuss the limitations of the proposed approach. It would be beneficial for the authors to include a brief discussion of potential limitations, for example from the perspective of the flat-map representation (e.g., possible geometric distortions or information loss) or from the benchmark construction aspect.

**Strengths And Weaknesses:**

Strength:
- The paper explores an interesting, intuitive and meaningful fMRI tokenization strategy based on cortical flat map projections, which, despite being conceptually straightforward, has not been systematically studied before. This provides a useful and practical exploration for adapting vision transformer architectures to fMRI data. The presentation is overall clear and well structured. The evaluation on the ablations and scaling performance is comprehensive.


Weakness:
- One concern is about the fairness of the comparison with previous baselines. The paper compares downstream performance with models like BrainJEPA and BrainLM using their released checkpoints, but the pretraining data is different. For example, BrainJEPA and BrainLM are pretrained on UK Biobank, while this work mainly uses HCP-YA. Even though the pretraining dataset here is smaller and the model still performs better on many tasks, it’s hard to tell whether the improvement comes from the proposed method or from differences in the pretraining data (e.g., dataset characteristics/quality). This makes the comparison a bit difficult to interpret.
- Also, since the model is pretrained on HCP-YA, I wonder whether the better performance on HCP-YA related downstream tasks (e.g., Sex prediction and Task21) might be partly due to domain overlap between the pretraining and evaluation data.
- Although this is a meaningful and worthwhile exploration, the technical contribution of the work feels a bit limited.

---

> ### Author Rebuttal · Authors · 2026-03-30
>
> Thank you for the helpful feedback. We appreciate the positive comments on the simplicity of our flat map approach, comprehensive experiments, and open benchmark.
>
> > BrainJEPA and BrainLM are pretrained on UK Biobank, while this work mainly uses HCP-YA.
>
> Great point. To test the effect of pretraining dataset, we now present an experiment where we pretrained CortexMAE on a subset of UKBB size-matched to our HCP-YA pretraining dataset (7.4M frames) as well as a larger subset consistent with BrainLM and Brain-JEPA (36M frames). We find UKBB pretraining results in modestly worse performance compared to HCP-YA. This is true both for HCP-YA Task21 (where there is no distribution shift for HCP-YA models), and NSD COCO24 (where there is similar distribution shift for all models).
>
> | Dataset | Frames | HCP-YA Task21 | NSD COCO24 |
> | --- | --- | ---: | ---: |
> | HCP-YA | 7.4M | 99.1 | 30.3 |
> | UKBB | 7.4M | 95.7 | 26.2 |
> | UKBB | 36M | 97.0| 26.8 |
>
> The weaker performance of UKBB pretraining is an interesting result to explore in future work. Nonetheless, the drop is small compared to the gap between our model and BrainLM/Brain-JEPA. Please see our response to Reviewer g5t6 for evidence that data normalization is instead the likely explanation.
>
> > the technical contribution of the work feels a bit limited.
>
> Since Reviewer g5t6 raises the same question, we provide our full response there
> .
> > In Table 4, the temporal patch size ablation shows that a patch size of 1 gives the best performance. I’m curious how the authors interpret this
>
> Our intuition is the same: temporal patch size reflects a speed/accuracy tradeoff. This is consistent with the results in [1] for classic ViTs. We chose not to adopt the smaller patch size for all other experiments because we felt the small performance gain didn’t justify the extra compute cost. We will clarify this choice and other rationale for the default base model in the final version.
>
> We speculate that smaller temporal patch size might be particularly important for dynamic state decoding under fast stimulus conditions. We only observe the effect on NSD, which uses fast 4s trials. We see no effect for static trait prediction, or for state prediction in slow block designs (HCP-YA). And we do not observe an improvement for smaller spatial patch size (8 vs 16). We will include these results in the final version.
>
> > Could the authors clarify the rationale behind the choice of the current downstream tasks
>
> Our choice of subject-level trait prediction tasks follows the recent work Brain-Harmony [2], which includes ABIDE, ADHD200, ADNI, PPMI diagnosis prediction as well as HCP-A. We chose not to include eval tasks based on UKB because the data are difficult to access and the usage agreement prohibits data sharing, making independent reproduction difficult. We will clarify these points.
>
> Our benchmark suite includes additional prediction targets for each dataset that are not currently reported in the paper. For example, below are more results for HCP-A. The overall message is consistent: trait prediction is variable and undifferentiated from the connectome baseline. Nonetheless, we agree these results are useful and will include them in the final version.
>
> | model | Fluid IQ | Crystal IQ | Memory | NEO-FFI N |
> |:---|---:|---:|---:|---:|
> | Connectome | 39.2 | 31.6 | 33.9 | 28.4 |
> | BrainLM | 35.6 | 27.2 | 35.9 | 25.5 |
> | Brain-JEPA | 24   | 23.6 | 23.2 | 27.2 |
> | CortexMAE  |  37.8 | 29.9 |  34.2 | 25.5 |
>
> Most importantly, our work includes an entire category of benchmarks that prior works ignore: dynamic state prediction (HCP-YA Task21 and NSD COCO24). These benchmarks are a crucial test of an fMRI FM’s sensitivity to genuine time-varying brain activity. Our finding that prior models perform poorly on these tests is, in our opinion, a necessary check on the field.
>
> > Converting 3D fMRI volumes into 2D cortical flat maps removes subcortical regions.
>
> The exclusion of subcortical structures is indeed a key limitation; Reviewer g5t6 raises the same question, we provide our full response there.
>
> > The paper does not explicitly discuss the limitations of the proposed approach.
>
> We apologize, yes there are many limitations: loss of subcortical structures, distortions due to flat map projection, information loss due to averaging over cortical thickness, homogenous HCP-YA pretraining data, etc. We will include a complete discussion of limitations in the final version.
>
> [1] Beyer, Lucas, et al. "Flexivit: One model for all patch sizes." CVPR 2023.
>
> [2] Dong, Zijian, et al. "Brain harmony: A multimodal foundation model unifying morphology and function into 1D tokens." NeurIPS 2025.

---

> > ### Author Rebuttal · Reviewer_8Ez3 · 2026-04-03
> >
> > Thank the authors for the rebuttal. My concerns have been largely addressed. Overall, I find this work to be a meaningful exploration with a valuable benchmark, and I will increase my score accordingly.

---

### Decision · Program_Chairs · 2026-04-30

**Decision:**

Accept (regular)

**Comment:**

This paper proposes an fMRI foundation model built on a simple but effective idea: projecting 3D fMRI volumes into 2D cortical flat maps so that standard spatiotemporal masked autoencoders and ViT-style architectures can be applied efficiently, while also providing a new benchmark for tokenization, scaling, and downstream evaluation in fMRI. Reviewers generally found the work technically solid, well executed, and valuable to the community, especially for its comprehensive empirical study, strong state decoding results, reproducible benchmark suite, and useful scaling analysis. Although several reviewers noted that the core methodological novelty is somewhat incremental and raised questions about fairness of some baseline comparisons, subcortical omission, and limited analysis of weaker clinical prediction results, the overall view was positive, and the rebuttal appears to have addressed most major concerns.